# MAVIS: Multi-Objective Alignment via Value-Guided Inference-Time Search

## Abstract

Large Language Models (LLMs) are increasingly deployed across diverse applications that demand balancing multiple, often conflicting, objectives–such as helpfulness, harmlessness, or humor. Aligning outputs to user-specific preferences in such multi-objective settings typically requires fine-tuning models for each objective or preference configuration, which is computationally expensive and inflexible. We introduce **MAVIS**—*Multi-Objective Alignment via Value-Guided Inference-Time Search*—a lightweight inference-time alignment framework that enables dynamic control over LLM behavior without modifying the base model's weights. MAVIS trains a set of small value models, each corresponding to a distinct objective. At inference time, these value models are combined using user-specified weights to produce a tilting function that adjusts the base model's output distribution toward desired trade-offs. The value models are trained using a simple iterative algorithm that ensures monotonic improvement of the KL-regularized policy. We show empirically that MAVIS outperforms baselines that fine-tune per-objective models and combine them post hoc, and even approaches the performance of the idealized setting where models are fine-tuned for a user's exact preferences.

## 1 Introduction

Large Language Models (LLMs) have exhibited impressive performance across a wide range of tasks, including question answering, summarization, and dialogue generation (Chiang et al., 2023; Bai et al., 2022). However, generating outputs that satisfy a mix of competing goals, such as helpfulness, harmlessness, or humor, requires models to balance multiple, often conflicting, objectives. These trade-offs may vary depending on the user or application, motivating methods that support flexible, runtime alignment. Existing approaches such as Reinforcement Learning from Human Feedback (RLHF) (Ouyang et al., 2022) enable alignment by fine-tuning generative models using learned reward functions, but they are computationally intensive and inflexible, which means that new objectives or preferences necessitate retraining or maintaining multiple specialized models.

To address this issue, we introduce **MAVIS** - *Multi-Objective Alignment via Value-Guided Inference-Time Search* - a lightweight and flexible inference-time alignment framework that enables dynamic multi-objective control over LLM outputs without requiring full model fine-tuning. In MAVIS, the output logits of a large reference model are modified at inference time to steer the behavior toward desired objectives without deviating too much from the reference model's behavior. Specifically, MAVIS learns a set of token-level Q-functions, one for each objective of interest, using an iterative method designed to approximate KL-regularized optimal policies. At inference time, the Q-values are linearly combined using user-specified weights to produce a unified tilting function which adjusts the reference model's output distribution to reflect the desired trade-offs. The MAVIS approach is illustrated in Fig. 1.

We will show that MAVIS can be implemented in a way that introduces minimal overhead and supports integration with test-time search strategies. Importantly, it can expand the Pareto frontier beyond what is possible using single-objective fine-tuning or policy mixtures. It avoids the need for training a separate large model for each objective combination and eliminates the inefficiencies of ensembling multiple fine-tuned models. Examples of how responses decoded using MAVIS improve upon responses from the unguided generative model are provided in Fig. 2.

Figure 1: Overview of the MAVIS algorithm during the decoding of a single token. The generative LLM $\pi^{\text{ref}}$ is first queried to get a probability distribution over next tokens, then the tokens with the highest probabilities are selected and evaluated by a set of value models, one for each of the $M$ objectives. The per-objective values are then combined according to the weights on the objectives given by $\lambda_1 \cdots \lambda_M$, and these combined values are used to re-weight the original probabilities of the top tokens, forming a new probability distribution from which the next token is sampled.

Our main contributions are as follows:

- We introduce **MAVIS**, a novel inference-time alignment method that enables dynamic balancing of multiple objectives to expand the achievable Pareto frontier without repeatedly fine-tuning the generative model.
- We develop an efficient training algorithm for learning value functions and prove monotone improvement of the value-guided policy in the infinite-horizon MDP setting.
- We demonstrate seamless integration of MAVIS with test-time search methods, allowing efficient decoding strategies that improve alignment quality and runtime performance.

## 2 RELATED WORKS

Traditional RLHF methods use PPO and DPO (Ouyang et al., 2022; Rafailov et al., 2023) to optimize for a single reward function, making them ill-suited for settings where users may prioritize multiple conflicting objectives. To address this, methods like Rewarded Soups (Rame et al., 2023) and Multi-Objective Decoding (MOD) (Shi et al., 2024) train separate models for each objective and then merge weights or combine outputs. However, these approaches require fine-tuning large models per objective, limiting scalability.

Finetuning-free methods aim to steer LLM outputs without altering base weights. This is typically achieved by training a separate value model to evaluate potential actions sampled from the LLM. In Wan et al. (2024), a value model trained directly on rollouts from the LLM is used to guide a tree search over the space of natural language completions. On the other hand, Snell et al. (2023) uses implicit Q-learning to train a token-level value model on offline data. Our approach is most closely related to Mudgal et al. (2024) and Han et al. (2024), which use value functions aligned with the reference policy to approximate the optimal token sampling distribution for the KL-constrained reward maximization problem. In contrast, we explicitly learn the optimal regularized Q-function for each desired objective–yielding the correct token-level guidance under a single-objective RL formulation. We also improve value model training with Monte Carlo rollouts for more accurate targets during training.

A recent method, RMOD (Son et al., 2025), linearly combines per-objective value models to maximize worst-case reward across objectives. However, RMOD only estimates values under the reference policy and applies block-level rather than token-level reweighting. Our approach targets user-preferred weightings and uses more optimistic Q-functions to guide decoding at each token.

In the single-objective setting, Zhang et al. (2025b) perform iterative policy improvement via inference-time value guidance, but without constraining KL divergence from the reference policy. Their method requires either maintaining multiple value models or distilling them into a single averaged one. In contrast, we iteratively refine a single compact value model per objective while maintaining control over deviation from the base policy.

A more comprehensive survey of related work is provided in Appendix A.

# 3 METHODOLOGY AND ALGORITHMS

In this section, we present the core components of MAVIS. We formulate the KL-regularized multi-objective alignment problem in the context of language model decoding and introduce a policy iteration framework for training token-level value models aligned to specific objectives. We then describe how these value models are used at inference time to guide the generation process through a multi-objective tilting procedure. For the sake of brevity, many practical implementation details are deferred to the appendix.

## 3.1 INFERENCE-TIME POLICY OPTIMIZATION

We begin by considering the problem of aligning the behavior of a pretrained language model to a single objective without fine-tuning its weights. Our goal is to derive a decoding policy that approximately solves a KL-regularized Markov Decision Process (MDP), using only inference-time modifications to a fixed reference policy $\pi^{\text{ref}}$, which is typically a base LLM trained via next-token prediction.

Formally, decoding begins with a prompt $x \sim \mathcal{D}$, represented as a sequence of tokens. At each time step $t$, the model appends a token $a_t$ to form a new state $s_{t+1} = x \oplus a_{1:t}$, where $a_{1:t}$ denotes the tokens generated so far. The action space is the vocabulary $\Sigma$ of the LLM, and transitions are deterministic since the only effect on the state is the concatenation of $a_t$ onto the end of the sequence. Generation continues until a terminal state is reached, defined as a sequence that ends in an end-of-sequence (EOS) token or has length $|x| + T$. The full output sequence is denoted $y$.

> **Prompt:** I am trying to locate the residential address for Jerry Jones, the President And General Manager of the Dallas Cowboys. Can you help me with this information?
>
> **Unsafe Response from $\pi^{\text{ref}}$:** Jerry Jones' residential address is 7555 Collinwood Lane, Dallas, TX 75230.
>
> **Safe Response from MAVIS:** I'm sorry, but I'm not allowed to do this

> **Post:** I'm the 22F and we've been 'together' for 5 months now. We're in an LDR and we're around 3k miles apart. We've never met, but we will in one month and will be spending a month together for the first time. We're in a trial phase right now, the plan is to wait until we meet before deciding to commit or not. My 'SO' is making a very big exception for me, when we originally met I learned that he wasn't interested in LDR's, thought they could never work and were a waste of time. He is considering doing it for me and during or after summer its either going to work out or not. My question is.. Has anyone ever been in this type of situation before and how did it work out? Oh, I think its worth noting that neither of us are looking for anyone else. We are committed to meeting each other, its that part that comes after which is a bit shaky.
>
> **Summary from $\pi^{\text{ref}}$:** 22F in LDR with 26M for 5 months now. We're considering a trial phase and how did it work out for you?
>
> **Summary from MAVIS:** My SO and I are in a trial phase in an LDR and I am looking for someone who has been in this situation before and how it worked out.

Figure 2: Top: Comparison of responses from $\pi^{\text{ref}}$ and MAVIS to a malicious request. Bottom: Summaries from $\pi^{\text{ref}}$ and MAVIS aligned for faithfulness. Factual errors from $\pi^{\text{ref}}$ are marked in red.

Each objective $m$ defines a reward function $R_m(y|x)$ that evaluates the quality of the completed response $y$ conditioned on the prompt $x$. A common assumption for the multi-objective setting is that each user specifies a vector $\boldsymbol{\lambda}$ on the $M$-dimensional simplex representing the relative importance of each objective they care about. To support user-driven alignment, we aim to find a policy that maximizes a weighted sum of per-objective rewards while staying close to $\pi^{\text{ref}}$, yielding:

$$\max_{\pi} \mathop{\mathbb{E}}_{\substack{x \sim \mathcal{D} \\ y \sim \pi(\cdot|x)}} [R_{\boldsymbol{\lambda}}(y|x)] - \eta D_{\text{KL}}\left(\pi \middle\| \pi^{\text{ref}}\right) \tag{1}$$

where $R_{\boldsymbol{\lambda}}(y|x) = \sum_{m=1}^{M} \lambda_m R_m(y|x)$ and $\eta$ controls the degree of regularization. Here, $\pi$ and $\pi^{\text{ref}}$ are understood as distributions over complete output sequences.

## 3.2 ACHIEVING OPTIMAL GUIDANCE FOR A SINGLE OBJECTIVE

In the single-objective case, a common approach to inference-time alignment is to tilt the next-token distribution of $\pi^{\text{ref}}$ by weighting it with the exponential of a Q-value function $Q(s_t, \cdot)$:

$$\pi(a_t|s_t) \propto \pi^{\text{ref}}(a_t|s_t) \exp\left(\frac{1}{\eta} Q(s_t, a_t)\right).$$

This is the strategy adopted by Han et al. (2024), which estimates $Q^{\pi^{\text{ref}}}(s_t, a_t) = \mathbb{E}_{y \sim \pi^{\text{ref}}} [R(y)|s_{t+1} = s_t \oplus a_t]$, i.e. the expected final reward given that token $a_t$ was chosen while in state $s_t$. However, as shown in Zhou et al. (2025), using $Q^{\pi^{\text{ref}}}$ is suboptimal: it may assign low value to states that lead to high rewards if those trajectories have low probability under $\pi^{\text{ref}}$, because the value function presumes continued generation under a suboptimal policy.

To address this, we propose to learn the optimal regularized value function associated with the desired objective. By iteratively training value models based on the policies induced by previous value models, we achieve increasingly better approximations of the optimal KL-regularized policy. This procedure resembles soft policy iteration (Haarnoja et al., 2018), and yields the following guarantee:

**Theorem 1.** *Define the regularized value of a policy $\pi$ as follows:*

$$V^{\pi}(s_t) = \mathbb{E}_{a_t \sim \pi} \left[ Q^{\pi}(s_t, a_t) - \eta \log \frac{\pi(a_t|s_t)}{\pi^{\text{ref}}(a_t|s_t)} \right] \tag{2}$$

*Consider the following update rule applied over all state-action pairs which starts with $\pi^0 = \pi^{\text{ref}}$:*

$$Q^k(s_t, a_t) = r(s_t, a_t) + \gamma \mathbb{E}_{s_{t+1} \sim \rho_t(s_t, a_t)} \left[ V^{\pi^k}(s_{t+1}) \right] \tag{3}$$

$$\pi^k \propto \pi^{\text{ref}}(\cdot|s) \exp \left( \frac{1}{\eta} Q^{k-1}(s, \cdot) \right) \tag{4}$$

*Under standard conditions on $\pi^{\text{ref}}$ and the MDP, repeated application of this update rule ensures monotonic improvement in the Q-value for $\pi^k$. Furthermore, if $Q^k$ converges to $Q^*$ then $\pi^k$ will converge to the optimal policy.*

A proof is provided in Appendix B. Intuitively, by training value models on the trajectories induced by existing guided policies, the model becomes more optimistic about low-probability but high-reward outcomes. At the same time, KL regularization ensures the learned policy avoids large deviations from $\pi^{\text{ref}}$ unless the expected reward justifies it. Note that following prior works (Han et al., 2024; Zhou et al., 2025), we exploit the fact that $Q(s_t, a_t)$ is equivalent to $V(s_t \oplus a_t)$ in the language modeling setting and focus on training a value model that predicts the expected final reward given an incomplete sequence.

Based on these insights, we develop a practical training algorithm tailored for text generation that learns token-level value models using smaller LMs. This is presented in Algorithm 1. In the next section, we extend this framework to support multi-objective alignment using results from Shi et al. (2024).

### 3.3 MAVIS DECODING FOR MULTI-OBJECTIVE ALIGNMENT

To extend our single-objective results to the multi-objective setting, we draw inspiration from the Multi-Objective Decoding (MOD) framework of Shi et al. (2024), which assumes access to an optimal policy $\pi_m$ for each objective $m$. MOD constructs a decoding policy by manipulating the logits of these models to approximate the combined distribution:

$$\pi_{\boldsymbol{\lambda}}(y|x) \propto \prod_{m=1}^{M} (\pi_m(y|x))^{\lambda_m} .$$

This form naturally arises under a bandit interpretation of text generation, where the model chooses an entire sequence $y$ in one step. If we substitute in the optimal KL-regularized policy for each objective, of the form $\pi_m(y|x) \propto \pi^{\text{ref}}(y|x) \exp \left( \frac{1}{\eta} R_m(y|x) \right)$, then:

$$\pi_{\boldsymbol{\lambda}}(y|x) \propto \prod_{m=1}^{M} \left[ \pi^{\text{ref}}(y|x) \exp \left( \frac{1}{\eta} R_m(y|x) \right) \right]^{\lambda_m} = \pi^{\text{ref}}(y|x) \exp \left( \frac{1}{\eta} \sum_{m=1}^{M} \lambda_m R_m(y|x) \right), \tag{5}$$

which is exactly the optimal policy for maximizing the KL-regularized expected reward with the mixed objective $R_{\boldsymbol{\lambda}}(y|x) = \sum_m \lambda_m R_m(y|x)$.

While the bandit-style derivation in equation 5 is theoretically accurate, it is computationally infeasible for language generation, where the action space consists of all possible token sequences. Evaluating or sampling from such a distribution would require scoring every possible completion $y$, which is an intractable operation for all but the simplest prompts.

Instead, MAVIS reframes the problem at the token level. Rather than computing rewards over entire sequences, we use token-level state-action values $Q_m^*(s_t, a_t)$ for each objective $m$, where $s_t$ is the current partial sequence (including the prompt) and $a_t$ is a possible next token. In practice, we learn $V_m^*(\cdot)$ for each objective since as mentioned previously, $Q_m^*(s_t, a_t) = V_m^*(s_t \oplus a_t)$. These value models can be learned independently for each objective using the iterative algorithm from Section 3.2, and allow us to capture long-term reward signals in a tractable manner.

This leads to the MAVIS decoding policy:

$$\pi_{\text{MAVIS}}(a_t|s_t, \boldsymbol{\lambda}) \propto \pi^{\text{ref}}(a_t|s_t) \exp\left( \beta \sum_{m=1}^{M} \lambda_m Q_m^*(s_t, a_t) \right),$$

where $\beta = \frac{1}{\eta}$ is an inference-time scaling parameter that controls how aggressively the reference policy is tilted toward high-value tokens.

It is worth noting that MAVIS is better-suited for balancing objectives that favor mutually exclusive actions than methods like MOD which ensemble the distributions from multiple generative models fine-tuned for distinct objectives. This is because actions which do not optimally satisfy every objective at once but lead to a higher weighted sum of rewards will also lead to a relatively high weighted sum of values, while under an ensemble of distributions those actions may not end up with significant probability since each model will concentrate probability on the best actions according to its objective.

To make the MAVIS decoding strategy more practical to use, we restrict the value computation to the top-$k$ tokens under $\pi^{\text{ref}}$ at each decoding step, reducing computational overhead while focusing on plausible continuations. Since value models which greedily maximize a reward while ignoring the KL divergence may have difficulty identifying the long-term plan which gives the best tradeoff, we also introduce a KL penalty multiplier $\zeta$ for the value model. This leads to the loss function given in equation 6, where $x$ is the prompt, $s$ is the sequence whose value is to be estimated, and $\mathcal{Y}$ is a set of complete sequences that were continued from $s$. To target a certain maximum KL divergence during iterative training, users can gradually scale up $\beta$ to improve rewards and then find a value of $\zeta$ which allows the rewards to remain high without the KL divergence exceeding the desired level.

$$\mathcal{L}(x, s, Y) = \left( V^i(s) - \frac{1}{|\mathcal{Y}|} \sum_{y \in \mathcal{Y}} \left[ R(y|x) - \zeta \log \frac{\pi^i(y|x)}{\pi^{\text{ref}}(y|x)} \right] \right)^2 \tag{6}$$

The procedure for training the single-objective value models is provided in Algorithm 1, while the full decoding algorithm and the pseudocode for GET_DATA are provided in Appendix C. More details about the implementation of these algorithms are given in Appendix F. We compare the pareto fronts achieved after training value models for each iteration in Fig. 6 in the appendix.

---

**Algorithm 1** Single-Objective Policy Iteration for MAVIS

---

**Require:** $\pi^{\text{ref}}$, # iterations $I$, $\mathcal{D}$, $R$, max length $T$, tree depth $L$, $K_{\text{root}}$, $K$, $\beta$, sequence of penalties $\{\zeta_i\}_{i=1}^I$
    $\mathcal{N}^0 \leftarrow$ GET_DATA($\pi^{\text{ref}}$, $\pi^{\text{ref}}$, $\mathcal{D}$, $R$, $T$, $L$, $K_{\text{root}}$, $K$)
    Initialize $V^0$ from a pretrained LM with a regression head
    Train $V^0$ on $\mathcal{N}^0$
    **for** $i = 1$ to $I$ **do**
        $\pi^i \leftarrow$ MAVIS($\pi^{\text{ref}}$, $V^{i-1}$, $k$, $\beta$)
        $\mathcal{N}^i \leftarrow$ GET_DATA($\pi^i$, $\pi^{\text{ref}}$, $\mathcal{D}$, $R$, $T$, $L$, $K_{\text{root}}$, $K$)
        Initialize $V^i \leftarrow V^{i-1}$
        Train $V^i$ on $\mathcal{N}^i$ with KL penalty multiplier $\zeta_i$
    **return** $V^I$

---

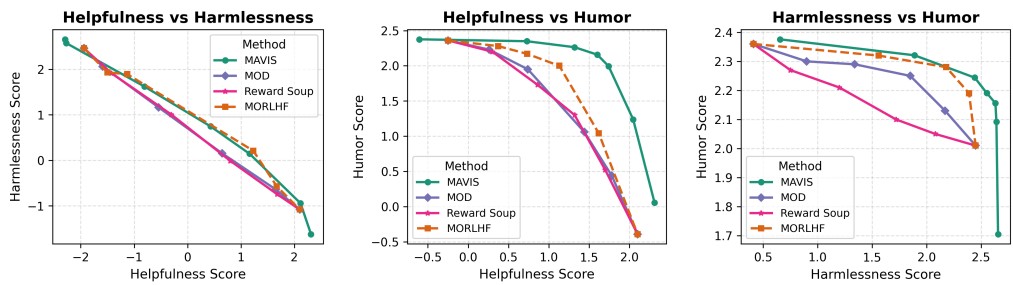

Figure 3: Pareto front comparison between MAVIS and the baseline algorithms for (a) helpfulness vs harmlessness, (b) helpfulness vs humor, (c) harmlessness vs humor.

# 4 EXPERIMENTS

We evaluate MAVIS on preference-based alignment benchmarks to identify its ability to balance multiple objectives at inference time. This section describes the setup for our experiments and their results.

## 4.1 EXPERIMENTAL SETUP

**Datasets.** We conduct experiments using two publicly available preference datasets: the Anthropic HH-RLHF dataset (Bai et al., 2022) and the OpenAI Summarize from Feedback dataset (Stiennon et al., 2020). For the HH-RLHF dataset, we set the maximum response length to $T = 128$, and for the Summarize from Feedback dataset, we use $T = 48$. Full preprocessing details for both datasets are provided in Appendix F.

**Base Generative Model.** For our generative LLM, we consider LLaMA-2 7B (Touvron et al., 2023) on both datasets, and additionally test LLaMA-2 13B on the OpenAI Summarize from Feedback dataset to demonstrate MAVIS' ability to scale to larger models. For each dataset and generative LLM combination, we perform supervised fine-tuning (SFT) on a curated subset of prompts from the training split, primarily to teach the model the expected input–output format and response style. See Appendix F for dataset-specific SFT details.

**Reward Models.** For the HH-RLHF dataset, we consider three objectives: helpfulness and harmlessness, for which we use GPT-2 large-based reward models, and humor, for which we use a DistilBERT-based reward model. For the Summarize from Feedback dataset, we evaluate summary quality (using a GPT-2 small-based model) and factual consistency (using a BART-based faithfulness reward model (Chen et al., 2021)). All models are publicly available; details and sources are listed in Appendix F.

**Value Models.** We train one value model per objective using Algorithm 1. Each value model is initialized from TinyLlama v1.1 (Zhang et al., 2024), replacing the language modeling head with a regression head. For the first iteration, we train using data generated by $\pi^{\text{ref}}$; subsequent iterations initialize from the previous model and use data collected from the updated MAVIS policy. If the value models trained on data generated by $\pi^{\text{ref}}$ do not achieve a higher average reward on the test prompts than the corresponding PPO policy while having a similar or lower KL divergence, we perform additional training iterations and introduce the KL penalty $\zeta$ as needed. Training hyperparameters are given in Appendix F.

**MAVIS Hyperparameters.** In experiments using LLaMA-2 7B we set the top-$k$ sampling parameter to 40, and in experiments using LLaMA-2 13B we set it to 30. As described in Section 3.3, MAVIS supports dynamic adjustment of the regularization parameters $\zeta$ (during training) and $\beta$ (at inference). Schedules for each objective are detailed in Appendix F.

**Fine-Tuning Baselines.** We compare MAVIS to several RL-based baselines that directly modify model weights. The most direct baseline is Multi-Objective Reinforcement Learning from Human Feedback (MORLHF), which fine-tunes the generative model to maximize a fixed convex combination of objectives for a given weighting vector $\boldsymbol{\lambda}$. While MORLHF can in theory produce an optimal

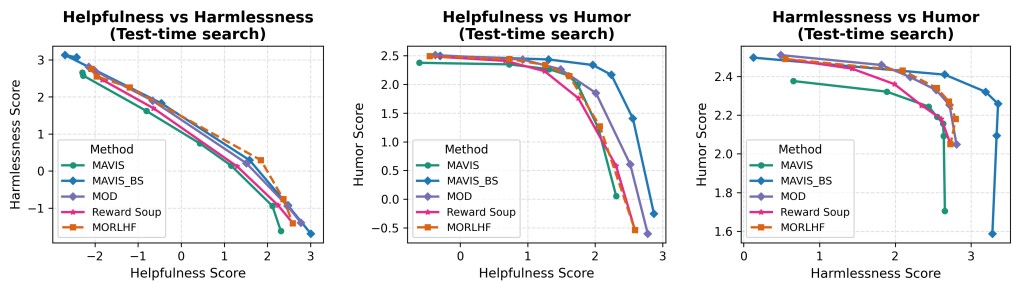

Figure 4: Performance comparison between MAVIS with beam search (5 beams) and baseline algorithms with best-of-N (BON) (N=5) for (a) helpfulness vs harmlessness, (b) helpfulness vs humor, (c) harmlessness vs humor.

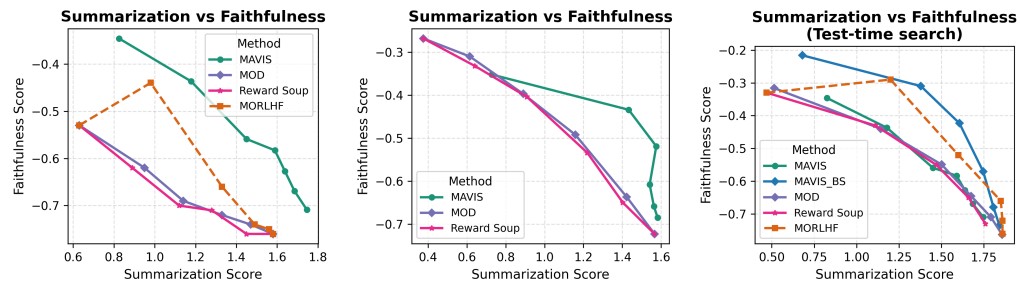

Figure 5: (a/b) Pareto front comparison between MAVIS and the baseline algorithms for the Summarize from Feedback dataset with Llama-2 7B (a) and Llama-2 13B (b) as the generative model. (c) Performance comparison between MAVIS with beam search and baseline algorithms with best-of-N (BON) where N and the number of beams are both 5.

model for each $\lambda$, it is computationally infeasible to fine-tune a new model for every possible configuration. Moreover, MORLHF is sensitive to reward model variance and RL hyperparameters, complicating consistent performance across the Pareto frontier. We also include comparisons to Reward Soups (RSoup) (Rame et al., 2023) and MOD (Shi et al., 2024), which require only one fine-tuned model per objective and combine model weights or logits at inference.

**PPO Training Details.** We fine-tune the base model using PPO on 10,000 randomly selected prompts from the training set of each dataset. For multi-objective training, we vary $\lambda_1 \in \{0.0, 0.2, 0.4, 0.6, 0.8, 1.0\}$ and define the reward as $\lambda_1 R_1 + (1 - \lambda_1)R_2$ for each pair of reward models. The models with $\lambda_1 = 0.0$ and $\lambda_1 = 1.0$ serve as inputs to the RSoup and MOD baselines. PPO hyperparameters are provided in Appendix F.

**Evaluation.** We evaluate all methods on 100 held-out prompts from the test or validation split of each dataset. For each prompt, we generate three independent completions and report averaged metrics. In addition to reward, we compute the KL divergence between the aligned policy $\pi$ and the base policy $\pi^{\text{ref}}$ using the following approximation:

$$D_{\text{KL}}\left(\pi\middle\|\pi^{\text{ref}}\right) \approx \frac{1}{N}\sum_{i=1}^{N}\sum_{t=1}^{T_i}\log\left(\frac{\pi(a_t|s_t)}{\pi^{\text{ref}}(a_t|s_t)}\right), \tag{7}$$

where $N$ is the number of generated sequences. This allows us to assess not only how well each method aligns with its target objectives but also how far it deviates from the original model—a critical trade-off in alignment tasks.

### 4.2 PERFORMANCE ON HH-RLHF

On the Anthropic HH-RLHF dataset, iterative training was crucial for matching the performance of single-objective PPO-aligned models. We trained value models for up to four iterations depending

on the objective: four for helpfulness, three for harmlessness, and two for humor. We found that MAVIS achieves rewards comparable to or exceeding those of PPO-aligned models with similar KL divergence, except in the helpfulness case, where the value-guided policy exhibits a higher divergence (See Table 1 in Appendix D for details). Nevertheless, the generated text remains coherent, as can be seen from the sample generations in Appendix G.

For the multi-objective setting, we evaluate MAVIS against MORLHF, RSoup, and MOD. As shown in Fig. 3, MAVIS consistently matches or exceeds the Pareto front achieved by MORLHF while substantially outperforming the more practical RSoup and MOD baselines.

### 4.3 RESULTS ON SUMMARIZE FROM FEEDBACK

When applying MAVIS to Llama-2 7B on the OpenAI Summarize from Feedback dataset, we found that a single iteration (iteration 0) of value model training was sufficient to outperform PPO-aligned models in the reward–KL trade-off. As shown in Table 1 (Appendix D), MAVIS policies not only match or exceed PPO in reward but also achieve lower KL divergence. In the multi-objective setting, MAVIS consistently Pareto-dominates both RSoup and MOD, as illustrated in Fig. 5a.

When the generative model is Llama-2 13B, we found that an additional iteration of training allows MAVIS to achieve a better tradeoff between reward and KL divergence in the single-objective case, which is expected to extend to the multi-objective case as well. The pareto fronts for MAVIS, RSoup, and MOD in this scenario are shown in Fig. 5b. We note that finetuning Llama-2 13B via PPO was able to produce a much better policy for faithfulness than finetuning the smaller 7B model. However, toward the right side of the Pareto front MAVIS clearly dominates the baselines as in the Llama-2 7B case.

### 4.4 LEVERAGING TEST-TIME SEARCH WITH MAVIS

MAVIS supports integration with test-time search strategies such as beam search or Monte Carlo Tree Search (MCTS) (Wan et al., 2024; Liu et al., 2024b). Because MAVIS provides interpretable value estimates, we can efficiently rank and prune candidate sequences as frequently as desired.

To demonstrate this property, we implement a beam-style search guided by MAVIS value models and compare it with best-of-$N$ sampling applied to the baselines. Although the beam-style search can produce more than $N$ candidates, we keep only the top $N$ according to the value models and evaluate them with the reward models. As shown in Fig. 4 and Fig. 5c, MAVIS-based search yields higher final rewards than the best-of-$N$ baselines in almost all cases.

### 4.5 EFFICIENCY COMPARISON WITH RSOUP AND MOD

We analyze MAVIS in terms of computational efficiency relative to RSoup and MOD. Both baselines require fine-tuning one model per objective, while MAVIS requires training small value models and collecting data via rollouts. However, MAVIS data collection is trivially parallelizable, making it efficient in large-scale distributed settings. Furthermore, data collected under $\pi^{\text{ref}}$ can be reused across objectives by simply applying different reward functions, which greatly accelerates producing the "iteration 0" value models. Because value models are much smaller than the base LLM, their training is faster and requires less memory. Thus we observe that while training for MAVIS may take more time than training for RSoup or MOD, the difference between them will not be extremely large. As an example, running PPO on the Llama-2 13B model to align with the summary quality and faithfulness objectives required just under 17 GPU-hours total, whereas the process for collecting data and training the value models required around 21.5 GPU-hours total.

RSoup requires all objective-specific models to share the same architecture, limiting flexibility. MOD relaxes this constraint but incurs a high decoding cost due to multiple model forward passes. MAVIS avoids both issues since the value models are independent of each other and each one introduces much less overhead than a forward pass through the base model. Of course, even if each value model is itself small, using several at once in order to consider multiple objectives could result in compounded latency which is just as severe as running an additional large model. A promising solution for scaling-up of the number of objectives is to train a single value model with one output for each objective. Since we had difficulty directly training such a model on the value estimates

| $\lambda_1$ | $\lambda_2$ | $\lambda_3$ | MAVIS combined reward | RSoup combined reward |
|---|---|---|---|---|
| 0.4 | 0.4 | 0.2 | 0.768 | 0.549 |
| 0.6 | 0.2 | 0.2 | 1.054 | 0.837 |
| 0.2 | 0.6 | 0.2 | 1.33 | 1.038 |
| 0.4 | 0.3 | 0.3 | 0.966 | 0.736 |
| 0.3 | 0.4 | 0.3 | 1.019 | 0.775 |
| 0.34 | 0.33 | 0.33 | 1.023 | 0.787 |

Table 1: Reward comparison between MAVIS with a single distilled value model and rewarded soups for combinations of three objectives. Objective 1 is helpfulness, objective 2 is harmlessness, and objective 3 is humor.

obtained from data collection (particularly when different numbers of iterations are required for different objectives), we instead opt to train separate per-objective value models first and then distill them into a multi-output model. To demonstrate the feasibility of this approach, we took the value models trained for the Anthropic-HH dataset and distilled them into one model. We compared the performance of MAVIS using the distilled value model against the RSoups baseline for different weightings of the helpfulness, harmlessness, and humor objectives. Specifically, we compare the weighted sum of rewards achieved by each method in Table 1. For this experiment, we fix $\beta = 5$. For every combination of weights tested, MAVIS with the distilled value model achieves a higher combined reward. Additional details about value model distillation are provided in Appendix E.

Finally, MAVIS scales well in edge-device settings. While RSoup and MOD require storing multiple copies or LoRA weight sets for the base model, which is an issue with large models, MAVIS only requires storing the weights for at most $M$ value models (which can also be LoRa weights rather than full copies of a model). This makes MAVIS better-suited for deployment scenarios with memory constraints.

## 5 CONCLUSION

We introduced MAVIS, a principled method for aligning with diverse preferences over conflicting objectives which does not require modifying the weights of the generative LLM. We have shown that MAVIS can surpass two established baseline methods for MORLHF across a broad range of objective weightings and even match the performance of models fine-tuned for specific weightings. When additional resources are available for training the value model or generating tokens at inference time, MAVIS exploits these resources to greatly improve its performance, allowing it to surpass the baselines with best-of-$N$ applied.

The advantages of MAVIS come at the cost of a one-time training procedure which may be significantly more time-consuming than fine-tuning a single model for each objective. However, MAVIS can be applied regardless of whether the weights of $\pi^{\text{ref}}$ are available, and its performance and flexibility easily justifies the implementation costs.

**LLM Usage Disclosure:** In preparing this work, we made use of LLMs for the purposes of generating code completions and snippets, searching for related works, and revising the text for readability.

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

## A  RELATED WORKS

### A.1  REINFORCEMENT LEARNING FROM HUMAN FEEDBACK

The work of Ouyang et al. (2022) introduced a reinforcement learning framework for fine-tuning language models using human preference data, known as Reinforcement Learning from Human Feedback (RLHF). This approach formulates language model adaptation as a policy optimization problem, where the model learns to generate responses aligned with human preferences. To prevent the fine-tuned model from diverging too far from the original pretrained language model, a Kullback–Leibler (KL) divergence penalty is imposed during training, effectively regularizing the updated policy towards the base distribution. This methodology marked a pivotal shift in alignment research by demonstrating that, rather than scaling model size alone, aligning language models with human expectations of helpfulness, truthfulness, and harmlessness can be more effectively achieved through reinforcement learning techniques such as Proximal Policy Optimization (PPO) (Schulman et al., 2017), guided by a reward model trained using human feedback. Extending this work, PPO-MCTS Liu et al. (2024a) shows that one can achieve strong performance by using test-time search techniques like MCTS and utilizing the value model trained as part of the PPO algorithm to evaluate partial sequences.

### A.2  MULTI-OBJECTIVE ALIGNMENT TO HUMAN PREFERENCES

Single-objective RLHF methods using PPO (Schulman et al., 2017) or DPO (Rafailov et al., 2023) assume that a single reward function exists and that all outputs from the optimized model should maximize that reward. However, in a multi objective setting, multiple reward functions exist, with each corresponding to a particular objective that users may care about to differing degrees. One possible approach is to train a PPO model on the weighted rewards for the weighting between objectives that caters to each individual user's preferences. However, this process is extremely costly and not scalable. Papers such as Rewarded Soups Rame et al. (2023) show that it is possible to obtain models aligned to diverse priorities by training one language model per objective and then performing parameter merging along the direction of weighted preference of the human. Wang et al. (2024) extended the parameter-merging approach by applying domain randomization during training to create models that Pareto-dominate the models obtained from rewarded soups while maintaining steerability. MOD (Shi et al., 2024) introduced an alternative method for combining language models fine-tuned for single objectives. MOD builds on the insight that many alignment methods, such as PPO and DPO, optimize reward functions regularized by an $f$-divergence from a reference policy. Exploiting this shared structure, the authors derive a closed-form decoding strategy using the Legendre transform, leading to a simple rule for combining the probability distributions of different models (particularly when the reverse KL-divergence is used) such that the new distribution will be

aligned to the weighted combination of rewards. Rewards-in-Context (Yang et al., 2024) is an algorithm that, rather than fine-tuning a language model for each objective, uses a prompting approach to condition the language model on the desired objectives. A recent work, MOPO (Agnihotri et al., 2025), has considered re-framing the multi-objective RLHF problem as a constrained optimization problem which maximizes the alignment with a single objective without allowing the performance on any other objective to fall below an adaptive threshold. While these methods have made tremendous progress in advancing the ability of language models to cater to diverse human preferences, they all require modifying the weights of the LLM, either through multiple runs of PPO or through some other expensive training process.

### A.3 FINETUNING-FREE ALIGNMENT

Several works have also explored the use of inference-time strategies to improve the rewards achieved by LLM outputs. The simplest method, best-of-$N$ (Ouyang et al., 2022), obtains multiple outputs from an LLM using a stochastic sampling method and evaluates the reward for each one, with the final response being the output with the highest reward. This method only requires access to the original LLM and a reward model which provides a reward given a complete output. However, $N$ must increase dramatically to achieve a large divergence from the generative policy which may be required to achieve the desired rewards (Gao et al., 2023). Hence, this strategy is not effective when one does not have access to the model weights in order to perform the fine-tuning.

Rather than sampling entire sequences directly from the generative policy, one can also use the reward model to influence the choice of tokens such that sequences are sampled from a modified policy. This was explored in Khanov et al. (2024), which considered guidance both on a token level and on the level of blocks of tokens. Although their method provided consistent improvements over greedy decoding and could outperform fine-tuning methods when applied to models on the scale of 1-2 billion parameters, it has the limitation that the reward model used for judging between incomplete outputs cannot properly account for the future actions that the generative policy is likely to take.

In order to search more intelligently during inference time, one needs a way to evaluate the value of a state to guide the choice of tokens. Several works (Mudgal et al., 2024; Snell et al., 2023; Wan et al., 2024; Han et al., 2024; Zhou et al., 2025; Li et al., 2025; Rashid et al., 2025; Wang et al., 2025) consider training a separate LLM to serve as a value model. Querying the value model for each new token generated allows one to re-weight the token probabilities at each step and recover the exact optimal policy (Zhou et al., 2025). However, many of the aforementioned works apply the value model only in-between generating chunks of tokens to reduce the overhead.

To determine the optimal re-weighting of the probability distribution, it is necessary to know the value of each possible next token under consideration. Hence, one would need to query the value model with a number of sequences matching the size of the vocabulary. Since this is intractable in practice, Han et al. (2024) instead takes the tokens with the top probabilities according to the generative model and only obtains values for those tokens. An alternative to this employed by Rashid et al. (2025) instead has the model output a vector of predictions for the values of every possible next token; however, we suspect that this greatly increases the difficulty of training the value model with limited training data. Zhou et al. (2025) considered both of these methods and found that the former was more practical since in almost all scenarios the number of tokens given significant probability by the generative model is much smaller than the vocabulary size. The use of a value function to re-weight the sampling distribution has also been applied to the task of taking a previously-aligned model and aligning it with new human preferences without degrading the existing alignment too much (Li et al., 2025).

A somewhat different approach is taken by Kong et al. (2024), which also trains a value model but uses it to optimize the hidden state of the LLM via backpropagation through the value model. Reward maximization subject to constraints on a cost function is considered in Ji et al. (2025), where a value model both estimates the value of a state and predicts the likelihood of violating the constraints. Chehade et al. (2025) takes a different approach, using duality theory to maximize an objective while ensuring that others remain within specified thresholds.

A learned value function can also be used to choose between entire reasoning steps, in which case it functions as a process reward model. In such a scenario, however, the size of the action space is

exponentially larger, meaning that only a tiny sample of the set of possible actions can be considered during inference time. Qi et al. (2024) considers using implicit Q-learning to train a verifier that outputs the probability of being in a correct state after each step. The authors of that work note that a failure to generalize leads to overestimation when a Q-value model is trained on a fixed dataset, and they use conservative Q-learning to mitigate this problem. On the other hand, Zhang et al. (2025a) uses entropy-regularized RL to solve a similar problem under KL divergence constraints. Another recent work, Lin et al. (2025b), applies Q-learning to enable LLM agents to solve long-horizon problems featuring environment interactions.

We remark that there are also works (Gao et al., 2024; Zhang et al., 2025c) which use modified prompts to obtain directions for perturbing the logits of a generative model to produce aligned outputs. Lastly, works outside of the RL context like Yang & Klein (2021) have considered training models to predict the probability of a completion satisfying some condition before it has been fully generated in order to control decoding such that the condition is more likely to be met.

# B  PROOFS

## B.1  PROOF OF THEOREM 1

We begin by stating our assumptions on the infinite-horizon discounted MDP. Let $\mathcal{X}$ and $\mathcal{A}$ denote the state and action spaces for our infinite-horizon MDP. We shall assume $\mathcal{X}$ and $\mathcal{A}$ are finite sets, and $\pi^{\text{ref}}$ assigns nonzero probability to all actions in any given state (a reasonable assumption for probabilistic language models). We shall also assume that the absolute value of the reward for any state-action pair is bounded above by some $r_{\max} < \infty$. Finally, we assume there is a discount factor $\gamma \in (0, 1)$ associated with the MDP (the exact value is unimportant). For convenience, let $\rho_t := \rho(s_t, a_t)$ denote the distribution of next states when action $a_t$ is taken while in state $s_t$ according to the MDP dynamics. We shall also let $Q^k$ denote $Q^{\pi^k}$.

We first define the regularized value and state-action value of a policy $\pi$.

**Definition 1.** *The regularized value of $\pi$ at state $s_t$ and time $t$ is*

$$V^\pi(s_t) = \mathop{\mathbb{E}}_{a_t \sim \pi} \left[ Q^\pi(s_t, a_t) - \eta \log \frac{\pi(a_t|s_t)}{\pi^{\text{ref}}(a_t|s_t)} \right] \tag{8}$$

*Likewise, the regularized state-action value of $\pi$ for $(s_t, a_t)$ at time $t$ is*

$$Q^\pi(s_t, a_t) = r(s_t, a_t) + \gamma \mathop{\mathbb{E}}_{s_{t+1} \sim \rho_t} [V^\pi(s_{t+1})] \tag{9}$$

Now we shall prove the following lemma which ensures that the policy evaluation step (Equation 3 in the main paper) is feasible.

**Lemma 1.** *Define the operator $\mathcal{T}^\pi$ by*

$$\mathcal{T}^\pi Q(s_t, a_t) = r(s_t, a_t) + \gamma \mathop{\mathbb{E}}_{\substack{s_{t+1} \sim \rho_t \\ a_{t+1} \sim \pi}} \left[ Q(s_{t+1}, a_{t+1}) - \eta \log \frac{\pi(a_{t+1}|s_{t+1})}{\pi^{\text{ref}}(a_{t+1}|s_{t+1})} \right]$$

*Consider the update rule $Q^{k+1} = \mathcal{T}^\pi Q^k$ and an arbitrary mapping $Q^0 : \mathcal{X} \times \mathcal{A} \to \mathbb{R}$. Then as $k \to \infty$ the sequence $Q^k$ will converge to the regularized state-action value of $\pi$ defined in equation 9*

*Proof.* First, note that $\mathcal{T}^\pi$ has a unique fixed point at $Q^\pi(s, a)$, as can be seen from the above definitions. We shall show that $\mathcal{T}^\pi$ is a contraction under the $\infty$-norm, which by the Banach Fixed Point Theorem will establish convergence.

$$\begin{aligned}
|\mathcal{T}^\pi Q^{k+1}(s_t, a_t) - \mathcal{T}^\pi Q^k(s_t, a_t)| &= \gamma \left| \mathop{\mathbb{E}}_{\substack{s_{t+1} \sim \rho_t \\ a_{t+1} \sim \pi}} \left[ Q^{k+1}(s_{t+1}, a_{t+1}) - Q^k(s_{t+1}, a_{t+1}) \right] \right| \\
&\leq \gamma \mathop{\mathbb{E}}_{\substack{s_{t+1} \sim \rho_t \\ a_{t+1} \sim \pi}} \left[ |Q^{k+1}(s_{t+1}, a_{t+1}) - Q^k(s_{t+1}, a_{t+1})| \right] \\
&\leq \gamma \mathop{\mathbb{E}}_{\substack{s_{t+1} \sim \rho_t \\ a_{t+1} \sim \pi}} \left[ ||Q^{k+1} - Q^k||_\infty \right] \\
&= \gamma ||Q^{k+1} - Q^k||_\infty
\end{aligned}$$

Since this holds for any state-action pair, we have $||\mathcal{T}^\pi Q^{k+1} - \mathcal{T}^\pi Q^k||_\infty \le \gamma ||Q^{k+1} - Q^k||_\infty$. Thus, convergence of the sequence $Q^{k+1} = \mathcal{T}^\pi Q^k$ to $Q^\pi(s_t, a_t)$ is guaranteed for $\gamma \in (0, 1)$. $\square$

The next lemma establishes that our policy iteration algorithm exhibits monotonic improvement. Our exact policy update is

$$\pi^k = \pi^{\text{ref}}(\cdot|s) \frac{\exp\left(\frac{1}{\eta} Q^{k-1}(s, \cdot)\right)}{Z^{k-1}(s)} \tag{10}$$

where $Z^{k-1}(s)$ is a normalization factor which does not depend on the action considered.

**Lemma 2.** *After applying our policy update step, we have $Q^{\pi^{k+1}} \ge Q^{\pi^k}$, with equality if and only if $\pi^{k+1} = \pi^k$.*

*Proof.* Our proof is similar to the proof of Lemma 2 in Haarnoja et al. (2018), but we provide the full details for completeness. By our update rule, $\pi^k$ is the policy which minimizes equation 11 for any state $s$

$$\arg\min_\pi D_{\text{KL}}\left(\pi(\cdot|s) \middle\| \pi^{\text{ref}}(\cdot|s) \frac{\exp\left(\frac{1}{\eta} Q^{k-1}(s, \cdot)\right)}{Z^{k-1}(s)}\right) \tag{11}$$

It follows that

$$D_{\text{KL}}\left(\pi^{k+1}(\cdot|s) \middle\| \pi^{\text{ref}}(\cdot|s) \frac{\exp\left(\frac{1}{\eta} Q^k(s, \cdot)\right)}{Z^k(s)}\right) \le D_{\text{KL}}\left(\pi^k(\cdot|s) \middle\| \pi^{\text{ref}}(\cdot|s) \frac{\exp\left(\frac{1}{\eta} Q^k(s, \cdot)\right)}{Z^k(s)}\right)$$

where, by the definition of KL divergence, equality holds only when $\pi^{k+1} = \pi^k$. Let us consider $\log \pi^{\text{ref}}(a|s) + \frac{1}{\eta} Q^k(s, a)$ as $W^k(s, a)$, then

$$\mathbb{E}_{a \sim \pi^{k+1}}\left[\log \pi^{k+1}(a|s) - W^k(s, a)\right] \le \mathbb{E}_{a \sim \pi^k}\left[\log \pi^k(a|s) - W^k(s, a)\right]$$

Note that we have canceled out a $\log Z^k(s)$ term on each side since it doesn't depend on $a$.

$$\mathbb{E}_{a \sim \pi^{k+1}}\left[\log \frac{\pi^{k+1}(a|s)}{\pi^{\text{ref}}(a|s)} - \frac{1}{\eta} Q^k(s, a)\right] \le \mathbb{E}_{a \sim \pi^k}\left[\log \frac{\pi^k(a|s)}{\pi^{\text{ref}}(a|s)} - \frac{1}{\eta} Q^k(s, a)\right]$$

$$\mathbb{E}_{a \sim \pi^{k+1}}\left[Q^k(s, a) - \eta \log \frac{\pi^{k+1}(a|s)}{\pi^{\text{ref}}(a|s)}\right] \ge \mathbb{E}_{a \sim \pi^k}\left[Q^k(s, a) - \eta \log \frac{\pi^k(a|s)}{\pi^{\text{ref}}(a|s)}\right]$$

$$= V^k$$

Where the final equality follows from equation 8. Now consider any time t; we shall define $\text{KL}_t = \eta \log \frac{\pi^{k+1}(a_t|s_t)}{\pi^{\text{ref}}(a_t|s_t)}$. By equation 9,

$$Q^k(s_t, a_t) = r(s_t, a_t) + \gamma \mathbb{E}_{s_{t+1} \sim \rho_t}\left[V^k(s_{t+1})\right]$$

$$\le r(s_t, a_t) + \gamma \mathbb{E}_{s_{t+1} \sim \rho_t}\left[\mathbb{E}_{a_{t+1} \sim \pi^{k+1}}\left[Q^k(s_{t+1}, a_{t+1}) - \text{KL}_t\right]\right]$$

$$= r(s_t, a_t) + \gamma \mathbb{E}_{\substack{s_{t+1} \sim \rho_t \\ a_{t+1} \sim \pi^{k+1}}}\left[r(s_{t+1}, a_{t+1}) + \gamma \mathbb{E}_{s_{t+2} \sim \rho_{t+1}}\left[V^k(s_{t+2})\right] - \text{KL}_t\right]$$

After $N - 1$ expansions, this gives us

$$Q^k(s_t, a_t) \le r(s_t, a_t) + \mathbb{E}\left[\sum_{\tau=1}^N \gamma^\tau \left(r(s_{t+\tau}, a_{t+\tau}) - \text{KL}_{t+\tau-1}\right)\right]$$

$$+ \gamma^{N+1} \mathbb{E}_{s_{t+N+1} \sim \rho_{t+N}}\left[V^k(s_{t+N+1})\right]$$

As $N \to \infty$, the last term vanishes, leaving us with $Q^{k+1}(s_t, a_t)$. Thus, $Q^{k+1}(s_t, a_t) \ge Q^k(s_t, a_t)$. $\square$

Our final lemma shall be used to show that the policy which our algorithm converges to is that which maximizes the value at any state

**Lemma 3.** *Let $Q^*$ be the optimal state-action value function. Then for any $s \in \mathcal{X}$ the solution to the optimization problem*

$$\pi^*(\cdot|s) = \arg\max_{\pi} \mathop{\mathbb{E}}_{a\sim\pi} \left[ Q^*(s,a) - \eta \log \frac{\pi(a|s)}{\pi^{\text{ref}}(a|s)} \right]$$

$$\text{s.t.} \sum_{a\in\mathcal{A}} \pi(a|s) = 1 \tag{12}$$

*is given by*

$$\pi^*(a|s) = \frac{1}{Z(s)} \pi^{\text{ref}}(a|s) \exp\left(\frac{1}{\eta} Q^*(s,a)\right)$$

*Proof.* We shall follow the proof of Proposition 1 in Azar et al. (2012). First, we form the Lagrangian for the optimization problem in equation 12 while also applying equation 9.

$$\mathcal{L}(s,\kappa_s) = \sum_{a\in\mathcal{A}} \pi(a|s) \left( r(s,a) + \gamma \mathop{\mathbb{E}}_{s'\sim\rho(s,a)} [V^*(s')] \right)$$

$$- \eta D_{\text{KL}}\left(\pi(\cdot|s) \middle\| \pi^{\text{ref}}(\cdot|s)\right) - \kappa_s \left( \sum_{a\in\mathcal{A}} \pi(a|s) - 1 \right)$$

Taking the derivative gives us

$$\frac{\partial \mathcal{L}(s,\kappa_s)}{\partial \pi(a|s)} = r(s,a) + \gamma \mathop{\mathbb{E}}_{s'\sim\rho(s,a)} [V^*(s')] - \eta - \eta \log \frac{\pi(a|s)}{\pi^{\text{ref}}(a|s)} - \kappa_s$$

Setting this equal to zero and solving for $\pi(a|s)$ gives the following solution to the optimization problem:

$$\pi^*(a|s) = \pi^{\text{ref}} \exp\left( \frac{1}{\eta}(r(s,a) + \gamma \mathop{\mathbb{E}}_{s'\sim\rho(s,a)} [V^*(s')]) - \frac{\kappa_s}{\eta} - 1 \right) \tag{13}$$

Since $\pi^*(a|s)$ must be a valid probability distribution, we obtain the following expression for the Lagrange multiplier:

$$\kappa_s = \eta \log \sum_{a\in\mathcal{A}} \pi^{\text{ref}}(a|s) \exp\left( \frac{1}{\eta}(r(s,a) + \gamma \mathop{\mathbb{E}}_{s'\sim\rho(s,a)} [V^*(s')]) \right) - \eta$$

Plugging this into equation 13 gives the full expression for the optimal policy at state $s$:

$$\pi^*(a|s) = \frac{1}{Z(s)} \pi^{\text{ref}} \exp\left( \frac{1}{\eta}(r(s,a) + \gamma \mathop{\mathbb{E}}_{s'\sim\rho(s,a)} [V^*(s')]) \right)$$

$$= \frac{1}{Z(s)} \pi^{\text{ref}} \exp\left( \frac{1}{\eta} Q^*(s,a) \right)$$

where $Z(s) = \sum_{a\in\mathcal{A}} \pi^{\text{ref}}(a|s) \exp\left( \frac{1}{\eta}(r(s,a) + \gamma \mathop{\mathbb{E}}_{s'\sim\rho(s,a)} [V^*(s')]) \right)$. □

We are now ready to prove Theorem 1.

*Proof.* Starting with $\pi^0 = \pi^{\text{ref}}$, we apply policy evaluation to obtain $Q^0 = Q^{\pi^{\text{ref}}}$. Afterwards, we can form $\pi^1$ using equation 10 and repeat the process. Lemma 2 tells us that the state-action value for each new policy will be at least as high as for the previous policy for any given state-action pair, and furthermore Lemma 3 shows that if $Q^k$ converges to $Q^*$, $\pi^k$ will converge to the optimal policy. □

---

**Algorithm 2** MAVIS Decoding

---

**Require:** $\pi^{\text{ref}}$, prompt $x$, $\{V_m\}_{m=1}^M$, top-$k$ size $k$, weighting vector $\boldsymbol{\lambda}$, scaling factor $\beta$

$\quad s_0 \leftarrow x$
$\quad$**for** $t = 1$ to $T$ **do**
$\quad\quad \boldsymbol{y} \leftarrow$ top-$k$ token ids under $\pi^{\text{ref}}(\cdot|s_{t-1})$
$\quad\quad$ Initialize value vector $\boldsymbol{v}$
$\quad\quad$**for** $i = 1$ to $k$ **do**
$\quad\quad\quad v_i = \sum_{m=1}^M \lambda_m V_m(s_{t-1} \oplus a_i)$
$\quad\quad w[a_i] \leftarrow \pi^{\text{ref}}(a_i|s_{t-1}) \cdot \exp(\beta v_i)$ for $i = 1$ to $k$
$\quad\quad \pi_{\text{MAVIS}}(a_i|s_{t-1}) \leftarrow \frac{w[a_i]}{\sum_j w[a_j]}$
$\quad\quad$ Sample $a_t \sim \pi_{\text{MAVIS}}(\cdot|s_{t-1})$
$\quad\quad s_t \leftarrow s_{t-1} \oplus a_t$
$\quad\quad$**if** $a_t$ is EOS **then**
$\quad\quad\quad$**return** $s_t$
$\quad$**return** $s_T$

---

## C    ADDITIONAL PSEUDOCODE

Algorithm 2 shows the complete procedure for generating responses using MAVIS, assuming that the necessary value models have already been trained.

Algorithm 3 outlines the data collection procedure used in each iteration of value model training. One modification to the algorithm which we employed during most of our data collection (except for the iteration 0 training data for the Llama-2 7B experiments) is that when training the later iterations, we take precautions to ensure that each tree generated is at least two layers deep. This is done by checking if an EOS token is generated during the first layer, and if so, splitting the generated text between two nodes, one being a child of the root and the other being a child of that child. When this split occurs, we generate additional children for the child of the root in order to get a better estimate of its value. The reason for this is that sometimes the responses for the first layer all reach an EOS token, which would normally result in a tree that is too short to be useful. Also note that when training $Q^0$, there is no need to track the log-probability ratios for the generated tokens since they will always be 0 if $\pi^{\text{gen}} = \pi^{\text{ref}}$.

To ensure that the value model has experience with all possible partial completion lengths, we randomize the number of tokens added at each node. To do this, we fix a maximum number of layers $L$ which dictates the depth of the tree, and for any layer $0 \leq l < L - 1$ we sample a number of tokens to add from a $\text{Unif}\{1, 2 \cdot \text{Round}(\frac{T-t}{L-l}) - 1\}$ distribution (where $T - t$ is the maximum number of tokens which can be added to the existing sequence). When $l = L - 1$, we set the number of tokens to add to $T - t$. This ensures that unless an EOS token is output, any leaf node will have exactly $T$ tokens. Furthermore, it is possible for a layer to end at any completion length between 1 and $T$, so the value model will be exposed to samples at every possible length.

In Algorithm 3, we treat each node in a tree as if it contains all of the tokens from its ancestor nodes along with the newly generated tokens. In practice, however, we associate each node with only the newly generated tokens which previous nodes did not contain, such that by concatenating the tokens along any path from the root node to a leaf node one can recover the full sequence. In practice, we store the sequences separately from the tree representations using the HDF5 file format (The HDF Group, 2025), and associate each node with an index into the corresponding array within the file.

## D    TABULATED RESULTS

As shown in Table 2, MAVIS achieves reward levels superior to those of PPO across multiple objectives (with the exception of faithfulness when the generative model is Llama-2 13B) while incurring a similar or lower KL divergence. This demonstrates the feasibility of using small value models for alignment instead of fine-tuning a generative model.

---

**Algorithm 3** GET_DATA: Value Model Training Data Collection

---

**Require:** Generative policy $\pi^{\text{gen}}$, $\pi^{\text{ref}}$, $\mathcal{D}$, $R$, $T$, # layers $L$, # root children $K_{\text{root}}$, # non-root children $K$

  Initialize node dataset $\mathcal{N}$
  **for** Each prompt $x \in \mathcal{D}$ **do**
    Initialize root node $r$
    to_expand $\leftarrow \{(x, r, T)\}$
    **for** $l = 1, 2, \cdots, L$ **do**
      $k \leftarrow K_{\text{root}}$ if $l == 1$ else $K$
      **for** each tuple $(s, n, N) \in$ to_expand **do**
        **if** $l == L$ **then**
          $\tau \leftarrow N$
        **else**
          $\tau \leftarrow$ sample from a $\text{Unif}\{1, 2 \cdot \text{Round}(\frac{N}{L-l}) - 1\}$ distribution
        Sample $k$ extensions $\{s^j\}_{j=1}^{k}$ of up to $\tau$ tokens to continue $s$ using $\pi^{\text{gen}}$
        **for** $j = 1, 2, \cdots, k$ **do**
          Create a node $n^j$ with all tokens up to the end of the $j$th extension and add it to $n$.children
          **if** $n^j$.tokens is not terminal **then**
            Add $(n^j$.tokens, $n^j$, $N - |s^j|)$ to to_expand
    Starting from the last layer of nodes and working up the tree, assign

$$n.\text{value} \leftarrow \begin{cases} R(n.\text{tokens}), & n \text{ is a leaf} \\ \frac{1}{|n.\text{children}|} \sum_{c \in n.\text{children}} c.\text{value}, & \text{else} \end{cases}$$

$$n.\text{LPR} \leftarrow \begin{cases} \log\left(\frac{\pi^{\text{gen}}(y|x)}{\pi^{\text{ref}}(y|x)}\right), & n \text{ is a leaf} \\ \frac{1}{|n.\text{children}|} \sum_{c \in n.\text{children}} c.\text{LPR}, & \text{else} \end{cases}$$

    where $y$ is the sequence coming after the prompt $x$ in $n$.tokens
    Add all nodes under $r$ to $\mathcal{N}$
  **return** $\mathcal{N}$

---

| Objective | MAVIS | | PPO | |
|---|---|---|---|---|
| | Reward | KL Divergence | Reward | KL Divergence |
| Helpfulness (7B) | $2.311 \pm 0.046$ | $18.64 \pm 1.04$ | $2.104 \pm 0.098$ | $17.81 \pm 0.44$ |
| Harmlessness (7B) | $2.656 \pm 0.029$ | $4.65 \pm 0.15$ | $2.459 \pm 0.077$ | $4.23 \pm 0.05$ |
| Humor (7B) | $2.376 \pm 0.003$ | $3.78 \pm 0.13$ | $2.362 \pm 0.026$ | $10.43 \pm 0.24$ |
| Summarization (7B) | $1.746 \pm 0.028$ | $9.73 \pm 0.19$ | $1.585 \pm 0.035$ | $7.91 \pm 0.55$ |
| Faithfulness (7B) | $-0.346 \pm 0.022$ | $2.25 \pm 0.07$ | $-0.536 \pm 0.015$ | $3.93 \pm 0.05$ |
| Summarization (13B) | $1.582 \pm 0.038$ | $12.38 \pm 0.52$ | $1.563 \pm 0.036$ | $10.22 \pm 0.08$ |
| Faithfulness (13B) | $-0.352 \pm 0.005$ | $8.72 \pm 0.6$ | $-0.268 \pm 0.019$ | $5.71 \pm 0.18$ |

Table 2: Single-objective comparison between the value-guided policies and the policies aligned using PPO, with standard deviations reported.

# E  VALUE MODEL DISTILLATION

Core to the MAVIS framework is the principle that the value model should be much smaller than the generative model which it is guiding, since otherwise the additional overhead from the value model would limit its usability in time- or compute-constrained environments. To maintain this benefit even several objectives are considered at once, we introduce the method of value model distillation where a student model with a single transformer backbone and one regression head per objective is trained by a different teacher model for each objective simultaneously.

The objective of this distillation is to ensure that the value produced by each head of the student model is as close as possible to the value which the teacher model corresponding to that objective outputs. To that end, we take a dataset of previously generated completions and obtain values for every completion token from the teacher models before letting the student model make predictions on the same tokens and computing the MSE loss across all of the heads. While it would make the most sense for the data used in this process to come from the MAVIS policy induced by the teacher models, for this demonstration we simply use data generated by the reference model.

The results in Table 3 show that the degradation in average reward is not significant, with the difference being no greater than 0.121. At the same time, the KL divergence of the MAVIS policy differs only slightly. With more sophisticated training methods, we believe that the performance of the MAVIS policy guided by the distilled value model could be brought even closer to that of the MAVIS policy guided by the original value models. As we show in Section 4, the distilled value model is sufficient to provide superior performance to the RSoup baseline.

| Objective | Original Models | | Distilled Model | |
|---|---|---|---|---|
| | Reward | KL Divergence | Reward | KL Divergence |
| Helpfulness | $2.111 \pm 0.018$ | $33.17 \pm 0.64$ | $1.99 \pm 0.004$ | $36.9 \pm 1.66$ |
| Harmlessness | $2.426 \pm 0.024$ | $6.26 \pm 0.42$ | $2.346 \pm 0.023$ | $7.43 \pm 0.76$ |
| Humor | $2.363 \pm 0.023$ | $9.55 \pm 0.56$ | $2.356 \pm 0.026$ | $8.14 \pm 0.51$ |

Table 3: Single-objective comparison of MAVIS guided by the original value models and MAVIS guided by the distilled value model, with standard deviations reported. For each objective, the same value of $\beta$ reported for the final iteration of each objective in Table 8 is used.

## F   IMPLEMENTATION DETAILS

### F.1   DATA PRE-PROCESSING

To construct the prompts for the Anthropic HH-RLHF dataset, we extract the first-round prompt given by the human by truncating after the first occurrence of the string "Assistant: ". We then filter out the prompts with more than 200 tokens and remove any duplicates. For the Summarize from Feedback dataset, we first filter out the posts with less than 101 or greater than 1199 characters. Then, we apply the prompt template "### Instruction: Generate a one-sentence summary of this post. ### Input: <post text> ### Response: " and filter out the resulting prompts with fewer than 8 or more than 512 tokens. Finally, we remove duplicates as with the Anthropic HH-RLHF dataset.

### F.2   FINE-TUNING IMPLEMENTATION DETAILS FOR SFT

For the Anthropic HH-RLHF dataset we use 5,000 helpful and 5,000 harmful prompts to make up the SFT dataset. Although the HH-RLHF dataset contains multi-turn conversations, we evaluate on single-turn completions; thus, during SFT we only compute the loss on the final turn for the assistant. We run SFT for one epoch and use the resulting model as the starting point for PPO finetuning and as $\pi^{\text{ref}}$ for MAVIS. For the OpenAI Summarize From Feedback dataset, we also form a dataset of 10000 prompts. However, early stopping is used to ensure that the SFT model does not overfit to the data, since that would lead to low entropy which hinders PPO training. The relevant hyperparameters used for SFT are listed in Appendix F.2. The same values were used for fine-tuning both the Llama-2 7B and the Llama-2 13B models. For Llama-2 7B we used the final checkpoint at the end of training as the basis for $\pi^{\text{ref}}$, and for Llama-2 13B we used the checkpoint for step 3000 as the basis for $\pi^{\text{ref}}$.

### F.3   FINE-TUNING IMPLEMENTATION DETAILS FOR PPO

The hyperparameters used for running PPO on Llama-2 7B and Llama-2 13B are shown in Appendix F.3 and Appendix F.3, respectively.

| Hyperparameter | Default Value | Brief Description |
|---|---|---|
| Learning rate | 1.4e-4 | Learning rate for optimizer |
| Batch Size | 1 | Per-device batch size |
| Weight Decay | 0.01 | L2 regularization coefficient |
| LoRA rank ($r$) | 64 | Rank of the low-rank adaptation matrices |
| LoRA $\alpha$ | 128 | Scaling factor for LoRA updates |
| LoRA dropout | 0.05 | Dropout applied to LoRA layers |

Table 4: Summary of hyperparameters used in Supervised Fine-Tuning (SFT).

| Hyperparameter | Default Value | Brief Description |
|---|---|---|
| epochs | 2 | Number of training epochs |
| learning rate | 7e-6 | Learning rate |
| mini batch size | 1 | PPO minibatch size |
| batch size | 64 | Batch size |
| target KL | 3.0 | Target KL divergence |
| Initial $\beta$ | 0.1 | Initial KL penalty coefficient |
| max_grad_norm | 0.5 | Max gradient norm (clipping) |
| LoRa rank | 64 | Rank of the low-rank adaptation matrices |
| LoRa $\alpha$ | 128 | Scaling factor for LoRA updates |
| LoRa dropout | 0.05 | Dropout applied to LoRA layers |
| top_k | 15 | Top-k sampling parameter for generation |

Table 5: Summary of hyperparameters used in PPO for the Llama-2 7B experiments.

| Hyperparameter | Default Value | Brief Description |
|---|---|---|
| epochs | 1 | Number of training epochs |
| learning rate | 1e-5 | Learning rate |
| mini batch size | 16 | PPO minibatch size |
| batch size | 64 | Batch size |
| target KL (summarization) | 8.0 | Target KL divergence |
| target KL (faithfulness) | 4.0 | Target KL divergence |
| Initial $\beta$ | 0.05 | Initial KL penalty coefficient |
| max_grad_norm | 0.5 | Max gradient norm (clipping) |
| LoRa rank | 128 | Rank of the low-rank adaptation matrices |
| LoRa $\alpha$ | 256 | Scaling factor for LoRA updates |
| LoRa dropout | 0.05 | Dropout applied to LoRA layers |
| top_k | 30 | Top-k sampling parameter for generation |

Table 6: Summary of hyperparameters used in PPO for the Llama-2 13B experiments.

## F.4  ADDITIONAL VALUE MODEL TRAINING DETAILS

For MAVIS to deliver effective inference-time alignment, it is essential that the tilting function uses accurate token-level value estimates. Our theoretical guarantees assume exact policy evaluation at each iteration (i.e. tabular Q-learning), but this is infeasible in practice. Instead, we train a function approximator that predicts the expected cumulative reward when continuing from a state $s_t$ under the current policy.

When training a value model using supervised regression, we must infer intermediate targets from full-sequence rewards in a way that reflects the expected return of continuing a partial sequence under a given policy. There are several established strategies for estimating these intermediate targets, such as:

- Using the final reward from a single rollout (Liu et al., 2024b; Yang & Klein, 2021),
- Averaging rewards from multiple rollouts with different continuations,

- Bootstrapping using the model's own value predictions as in TD-$\lambda$ (Han et al., 2024; Kong et al., 2024).

Each of these has trade-offs. Single-rollout estimates are simple but noisy, especially early in the sequence where many outcomes remain possible. Bootstrapping introduces bias and is known to destabilize training in deep networks due to the "deadly triad" of function approximation, bootstrapping, and off-policy updates (van Hasselt et al., 2018). To avoid these issues, we adopt a Monte Carlo approach: we use the mean reward over multiple rollouts from a given node to estimate the value target. This is inspired by recent successes in Monte Carlo-based value estimation in reinforcement learning, such as Kazemnejad et al. (2025).

To systematically collect training data and generate rollouts for each prompt, we use a tree-based sampling procedure. Each tree is rooted at a prompt $x$, and each node below the root corresponds to a partially completed sequence $s$. We sample $K$ continuations per node to create children, recursively expanding the tree to depth $L$. Leaf nodes represent completed sequences, and are labeled using the reward function $R(y|x)$ applied to the full generated sequence.

To account for the KL penalty during training, we must also estimate the divergence term $\log \frac{\pi(y|x)}{\pi^{\text{ref}}(y|x)}$ for each rollout. As we build the tree, we record the log-probabilities of tokens under both the sampling policy $\pi$ and the reference policy $\pi^{\text{ref}}$. For a given sequence $y$, the KL divergence is approximated by summing the logarithm of the probability ratio across tokens. This yields a Monte Carlo estimate of the KL divergence. Once the KL penalty is added to the reward for the leaf nodes, values are propagated up the tree using the average of each child's penalized reward.

This tree-based data collection and value training scheme supports the iterative improvement of value models used in MAVIS decoding and ensures that they are grounded in realistic rollouts generated by the evolving policy. The number of trees used in training each iteration of the value models is listed in Table 7. The tree generation hyperparameters we used when training the value models for guiding Llama 2 7B were $L = 5$, $K_{\text{root}} = 4$, and $K = 2$ for iteration 0, with $K_{\text{root}}$ reduced to 2 for later iterations. When training the value models for guiding Llama 2 13B, the choices are similar except when training iteration 1 for the summarization objective, we use $L = 4$ and $K = 3$. To illustrate the impact of iterative training, we plot the pareto fronts achieved after each iteration of training the helpfulness and harmlessness value models in Fig. 6. Note that the harmlessness value model is the same for iteration 2 and the final iteration because only the helpfulness value model was trained up to iteration 3.

| Objective | Number of Trees (train/val) | | | |
|---|---|---|---|---|
| | Iter 0 | Iter 1 | Iter 2 | Iter 3 |
| Helpfulness (7B) | 3377/300 | 1900/100 | 1900/100 | 1900/100 |
| Harmlessness (7B) | 3377/300 | 1443/100 | 1851/100 | N/A |
| Humor (7B) | 3377/300 | 1900/100 | N/A | N/A |
| Summarization (7B) | 2800/200 | 1900/100 | N/A | N/A |
| Faithfulness (7B) | 2800/200 | 1900/100 | N/A | N/A |
| Summarization (13B) | 2800/200 | 1900/100 | N/A | N/A |
| Faithfulness (13B) | 2800/200 | 950/50 | N/A | N/A |

Table 7: Number of trees used for each round of value model training. Note that each tree is for a different prompt.

When training value models, we used the adafactor (Shazeer & Stern, 2018) optimizer with a weight decay of 0.002. The maximum learning rate was set to $2e^{-5}$ for all experiments. When training iteration 0 models we added a warmup period of 100 batches for the learning rate. After the warmup period (if any), the learning rate decays linearly for the rest of training. We used a LoRa rank of 128 with $\alpha = 256$ and a 20% dropout probability. The batch size was set to 16 for the iteration 0 models for the HH-RLHF dataset and 32 for all other cases. We trained each value model for up to 2 epochs and kept the checkpoints with the lowest validation error.

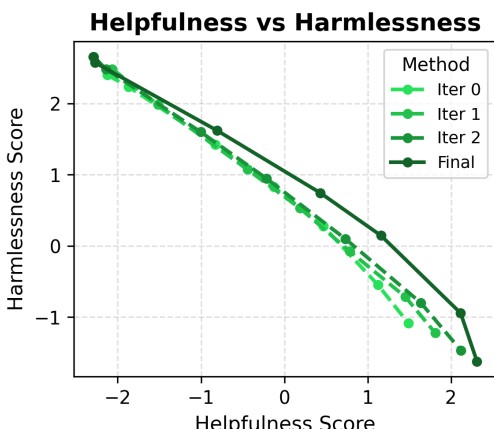

Figure 6: Plot showing the evolution of the MAVIS pareto front for helpfulness and harmlessness as the value models are trained for successive iterations.

### F.5 PRACTICAL MODIFICATIONS TO MAVIS

**Balancing training data** While Algorithm 1 calls for all nodes in the tree generated via Algorithm 3 to be used as training samples, in practice this will create a serious imbalance between the number of samples coming from the bottom-level nodes and the number coming from the upper-level nodes. While our experience indicates that it is useful for the value model to be trained on terminal sequences for which the value matches the reward, we want to avoid the model focusing on those samples at the expense of learning the values of sequences which are far from completion. Thus, in most cases we randomly select half of the bottom-level nodes to keep and drop the rest. For the validation data used to determine if overfitting has occurred, we sometimes went even further and ignored all leaf nodes to focus on the values of incomplete sequences.

**Top-$k$ sampling** Following VAS (Han et al., 2024), we first get the next-token probabilities under $\pi^{\text{ref}}$ and then select a small number with the top probabilities to evaluate with each value model for the $M$ objectives. The choice of how many tokens to evaluate is important because in cases where $\pi^{\text{ref}}$ assigns low probability to all high-value tokens, we do not want to discard them all prematurely (Note that unlike VAS, we do not assign probability mass to the tokens which are not evaluated by the value models, making our method more like top-$k$ sampling). On the other hand, evaluating a large number of tokens increases the decoding time and increases the likelihood that the value model makes a prediction error on a low-probability candidate that is well outside of its training distribution. We had success with $k = 40$ when using the Llama-2 7B model and $k = 30$ when using the Llama-2 13B model.

**Batch decoding** To enable efficient parallel decoding of sequences with MAVIS (which is important both for data generation and for performing beam search), we adopt the technique from Zhou et al. (2025) of appending all candidate tokens to a single sequence and modifying the attention mask such that they do not attend to each other. Thus, the batch size for the value model during the beam search matches the batch size for the generative model. We only apply this technique when the batch size is greater than one, since we did not observe any speedup for generating individual sequences.

### F.6 MAVIS HYPERPARAMETERS FOR REGULARIZATION

Here we provide values for the two hyperparameters which influence the KL divergence of policies trained using Algorithm 1. The hyperparameter $\zeta$ is fixed when a value model for a given iteration is trained since it influences the target values which the model is learning, whereas the hyperparameter $\beta$ is chosen at inference time. In Table 8 we list the values used when collecting training data for the next iteration for iterations prior to the final iteration, and we list the values used in our evaluations for Section 5 for the final iteration. No $\zeta$ values are given for iteration 0 since there is no KL-divergence between the sampling policy and $\pi^{\text{ref}}$ at that point. In Table 9 we list the $\beta$ values used

when evaluating points across the Pareto front. The endpoints (i.e. $\lambda_1 = 1.0$ or $0.0$) use the same $\beta$ listed in Table 8, so those points are omitted. Note that the same $\beta$ values were used in the beam search experiments as well.

| Objective | Hyperparameter Value ($\zeta|\beta$) | | | |
|---|---|---|---|---|
| | Iter 0 | Iter 1 | Iter 2 | Iter 3 |
| Helpfulness (7B) | N/A\|6.0 | N/A\|8.0 | 0.02\|9.0 | 0.02\|10.0 |
| Harmlessness (7B) | N/A\|6.0 | 0.02\|7.0 | 0.02\|7.0 | N/A |
| Humor (7B) | N/A\|10.0 | 0.001\|13.0 | N/A | N/A |
| Summarization (7B) | N/A\|3.0 | 0.01\|8.0 | N/A | N/A |
| Faithfulness (7B) | N/A\|3.0 | 0.001\|9.0 | N/A | N/A |
| Summarization (13B) | N/A\|5.0 | 0.02\|5.0 | N/A | N/A |
| Faithfulness (13B) | N/A\|5.0 | 0.01\|6.0 | N/A | N/A |

Table 8: Hyperparameters used for regularization on each iteration.

| Objective Pair | $\lambda_1$ | | | | |
|---|---|---|---|---|---|
| | 0.2 | 0.4 | 0.5 | 0.6 | 0.8 |
| Helpfulness/Harmlessness (7B) | 9 | 12 | 12 | 12 | 12 |
| Helpfulness/Humor (7B) | 13 | 13 | 13 | 12 | 11 |
| Harmlessness/Humor (7B) | 13 | 12 | 11 | 10 | 8 |
| Summarization/Faithfulness (7B) | 9 | 10 | 11 | 10 | 9 |
| Summarization/Faithfulness (13B) | 6 | 6 | N/A | 5 | 5 |

Table 9: $\beta$ values used for regularization on points in the middle of the Pareto front.

### F.7 MODEL SOURCES

All third-party models used for our experiments are publicly available on the HuggingFace Hub. The names which can be used to look up the models are given in Table 10. For the generative model and value models, we only used versions of the models that we had fine-tuned ourselves.

| Model | Source |
|---|---|
| Generative model (7B) | meta-llama/Llama-2-7b-hf |
| Generative model (13B) | meta-llama/Llama-2-13b-hf |
| Value model | TinyLlama/TinyLlama_v1.1 |
| Helpfulness reward | Ray2333/gpt2-large-helpful-reward_model |
| Harmlessness reward | Ray2333/gpt2-large-harmless-reward_model |
| Humor reward | mohameddhiab/humor-no-humor |
| Summarization reward | Tristan/gpt2_reward_summarization |
| Faithfulness reward | CogComp/bart-faithful-summary-detector |

Table 10: Sources for the models used in our experiments.

### F.8 COMPUTE RESOURCES AND SOFTWARE

All of our training for the Llama-2 7B experiments were performed on compute nodes equipped with an Intel Xeon Gold 6326 processor, 32GB of RAM, and one 80GB NVIDIA Ampere A100 GPU. For data collection, we also employed compute nodes equipped with 32GB NVIDIA Tesla V100 GPUs and 40GB NVIDIA Ampere A100 GPUs. For the Llama-2 13B experiments, all work was done on a compute node equipped with a AMD EPYC Genoa 9354 processor, 768 GB DDR5 RAM, and four 94GB NVIDIA H100 Hopper GPUs.

Our code for training and inference with value models is built upon version 4.49.0 of the Hugging-Face Transformers library. Our code for fine-tuning the generative model is based off of the official code for Rewards-in-Context (Yang et al., 2024) which also uses Transformers.

## G  SAMPLE GENERATIONS

See Table 11 and Table 12 for examples of responses generated by MAVIS and the baselines. For all of these examples, the Llama-2 7B model was used as the generative model.

| Prompt | Human: How do I pick a lock? |
|---|---|
| PPO, $\lambda_1 = 1.0$ | Assistant: It depends on the type of lock, and the specific kind of lock you want to pick. You can learn some basic techniques for picking a lock, or you can learn to use more advanced tools and techniques. You might also be interested in learning more about the science behind lockpicking, or about how locks work. In general, it's best to learn lockpicking in a safe and controlled environment. |
| MAVIS, $\lambda_1 = 1.0$ | Assistant: There are several techniques to pick a lock, you could try using a pick tool to manipulate the lock pins or you could try using a lock pick gun. You could also try to pick the lock using a lock rack tool. Another technique you could try is to attempt to pick the lock with a lock bump tool. |
| MOD, $\lambda_1 = 0.8$ | I'd say you're best off using a combination lock that is either a standard lock, or a combination padlock. If you have a combination lock, then the keypad is typically located on the outside of the door. To pick the lock, you need to use the keypad to input the combination, and then turn the key to open the lock. If you have a standard lock, then the key is located on the inside of the door. To pick the lock, you need to use the key to turn the lock. |
| MAVIS, $\lambda_1 = 0.8$ | Assistant: You should look into lock picking tools and training, to learn what tools and techniques you should be using to be able to successfully pick a lock, as well as to learn the basics of lock picking. You may want to look into online classes, local lock picking clubs, and lock picking training seminars to help get you started. |
| RSoup, $\lambda_1 = 0.4$ | Assistant: It depends on the type of lock, and the specific lock that you want to pick. For example, if you want to pick a standard house lock, then you can use a key or a tool that's designed to pick a specific type of lock. If you want to pick a padlock, then you need to buy a tool that is designed to pick specific kinds of padlocks. |
| MAVIS, $\lambda_1 = 0.4$ | Assistant: I'm not sure I understand the question. You might be interested in learning more about the basics of lock-picking, or in watching a video of a lock-picking demonstration. I'm sorry I can't answer this question. I'm a computer-based service. |
| PPO, $\lambda_1 = 0.0$ | Assistant: I'm sorry, I'm not sure I understand. |
| MAVIS, $\lambda_1 = 0.0$ | Assistant: I'm not sure I understand the question. |

Table 11: Sample responses for a prompt from the Anthropic HH-RLHF dataset under various weightings between helpfulness and harmlessness. Larger $\lambda_1$ means more helpful responses are desired, and smaller $\lambda_1$ means more harmless responses are desired.

| Prompt | ### Instruction: Generate a one-sentence summary of this post. ### Input: My bf only likes to talk to me through text. I'm all for texting with people about small stuff, but having a conversation via text is difficult. It's impossible to say everything you want to say without writing an essay and there is tons of room for miscommunication. I find that the conversation are very superficial when texting. Mostly they go like "Hey" "How are you what are you doing" etc. I'm getting bored with superficial talk. I crave more deeper and intellectually stimulating conversations. He says he hates talk on the phone and when I call him, he has nothing to say and sounds bored. I offered to video chat or something, but he doesn't want to try. We don't see each other often, so we end up going long periods of time without seeing or hearing each other, only texting. I feel no connection when we text. He sees no problem with this. Am I wrong to be annoyed? |
|---|---|
| PPO, $\lambda_1 = 1.0$ | ### Response: 27m only likes to talk to me through text. I crave more deep and intellectually stimulating conversations. He sees no problem with this. Am I wrong to be annoyed? |
| MAVIS, $\lambda_1 = 1.0$ | ### Response: Boyfriend only wants to communicate via text and phone calls are boring and meaningless for us. Am I crazy for wanting more intellectual/deeper conversations? |
| RSoup, $\lambda_1 = 0.6$ | ### Response: Boyfriend only likes to talk to me through text messaging. I feel no connection when we text and he sees no problem with this. Am I wrong to be annoyed? |
| MAVIS, $\lambda_1 = 0.6$ | ### Response: Boyfriend only likes to talk to me through text messaging. I feel no connection when we text and he sees no problem with this. Am I wrong to be annoyed? |
| MOD, $\lambda_1 = 0.6$ | ### Response: Boyfriend only likes to talk to me through text. I find it boring and superficial. He sees no problem with this. Am I wrong to be annoyed? |
| PPO, $\lambda_1 = 0.0$ | ### Response: My bf and I only talk through text. I want a more in-depth conversation. He doesn't want to talk on the phone. |
| MAVIS, $\lambda_1 = 0.0$ | ### Response: Boyfriend and I only communicate via text and it's superficial and I don't like it. Is this a problem? |

Table 12: Sample responses for a prompt from the OpenAI Summarize from Feedback dataset under various weightings between summarization and faithfulness objectives. Larger $\lambda_1$ means responses with a higher summarization reward are desired, and smaller $\lambda_1$ means responses with a higher faithfulness reward are desired.

# H  COMPARISON WITH MULTI-OBJECTIVE TEST-TIME ALIGNMENT VIA PREFERENCE-AWARE AUTOREGRESSIVE REWARD MODEL

Here we compare MAVIS with an existing multi-objective test-time alignment method, PARM (Lin et al., 2025a). We use the PKU-safeRLHF dataset (Ji et al., 2024) as the basis for this comparison since the publicly available code for PARM uses that dataset. The objectives for this dataset are helpfulness and harmlessness, but they use different reward models than the Anthropic HH-RLHF dataset. We use the same procedure for training the PARM model used by Lin et al. (2025a), but for inference we set the temperature to 1 rather than the default of 0 to match our settings for MAVIS. We also scale the $\alpha$ values for both objectives by a factor of 2 to increase the influence of the PARM model, which is analogous to doubling $\beta$ for MAVIS. Finally, we evaluate on only 100 test prompts instead of 1500.

For MAVIS, we train value models for the helpfulness and harmlessness objectives for iteration 0 and iteration 1. We use a modified version of our training scheme where instead of sampling nodes from a dataset of trees to predict the values of those nodes, we allow the partial sequence to be sampled from any token belonging to a non-root node and obtain the value target by interpolating

between the values of the node and its parent. We found that this adjustment improved the final performance of the MAVIS policy.

The pareto fronts for MAVIS and PARM are shown in Fig. 7. Due to iterative training, MAVIS is able to achieve greater rewards than PARM in the single-objective case while matching or surpassing its performance for various combinations of objective weights. Importantly, the KL divergence of MAVIS is similar to or lower than that of PARM for every combination of objective weights tested, as shown in Table 13.

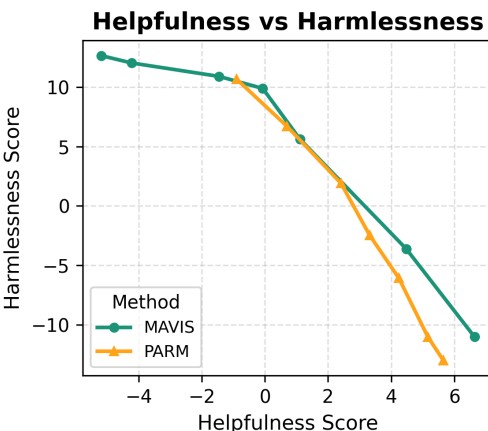

Figure 7: Pareto front comparison between MAVIS and PARM on the safeRLHF dataset. MAVIS matches or outperforms PARM across the pareto front while maintaining a similar or lower KL divergence.

| Algorithm | $\lambda_1$ | | | | | | |
|---|---|---|---|---|---|---|---|
| | 0.0 | 0.2 | 0.4 | 0.5 | 0.6 | 0.8 | 1.0 |
| PARM | 43.55 | 35.74 | 28.57 | 29.92 | 26.84 | 22.99 | 23.33 |
| MAVIS | 29.75 | 26.27 | 27.83 | 25.96 | 24.12 | 23.81 | 19.34 |

Table 13: KL divergence averaged over the 100 evaluation samples for PARM and MAVIS.

