# OpenReview forum: "MAVIS: Multi-Objective Alignment via Value-Guided Inference-Time Search"
_ICLR.cc/2026/Conference — Submitted to ICLR 2026_

### Official Review · Reviewer_8deh · 2025-10-28

**Soundness:** 2
**Presentation:** 2
**Contribution:** 2
**Rating:** 2
**Confidence:** 3

**Summary:**

The paper proposes MAVIS, a value-guided decoding framework that enables a plug-and-play framework for multi-objective alignment. The main technical contribution involves resembling soft policy iteration to better estimate the Q function.

**Strengths:**

- LLM multi-objective alignment is important.

- The proposed method is overall reasonable.

**Weaknesses:**

- The presentation of this paper requires further refinement. Certain passages are overly informal; for instance, line 431 begins with “Of course…”. Additionally, some concepts are inadequately explained—for example, ζ is not properly defined in Section 3. Figure 2 in the methodology section seems unnecessary. Overall, the paper is a little hard to follow, especially for RL non-experts.

- The motivation of the work is unclear. Previous methods such as GenARM [1] and PARM [2] have already explored test-time multi-objective alignment in LLMs.

- The novelty of the paper is limited. The core contribution revolves around soft policy iteration, which has been explored in prior works. No fundamentally new algorithm is introduced.

- The current comparison set is narrow. The authors should include more comprehensive baselines: (1) For training-based methods: compare against MODPO, RiC, and other SOTA methods. (2) Compare against more methods of model merging. (3) Directly compare with other multi-objective decoding methods such as GenARM and PARM.

- The test set contains only 100 held-out prompts, which may limit robustness. Consider using benchmarks such as AlpacaEval 2 which leverage LLM-judge frameworks to assess helpfulness and alignment more reliably.

- The paper lacks quantitative analysis of the impact of guidance on generation quality, such as fluency and diversity.

- Lack of more analysis: (1) the role of Top-K sampling; (2) the performance change during each iteration when training value functions; (3) the cost analysis to estimate value functions.

- The authors claim that “MAVIS can be applied regardless of whether the weights of π_ref are available.” However, it is unclear how distribution adjustment can be performed without access to these weights.

- The process of training value functions may require more time compared to directly training the policy. Additionally, the training stability remains uncertain, and the performance under long output sequences has not been explored.

Reference

[1] GENARM: REWARD GUIDED GENERATION WITH AUTOREGRESSIVE REWARD MODEL FOR TEST-TIME
ALIGNMENT

[2] PARM: Multiobjective test-time alignment via preference-aware autoregressive reward model.

**Questions:**

See weaknessness.

---

> ### Author Response · Authors · 2025-11-21
> **Author Rebuttal**
>
> "The presentation of this paper requires further refinement...": Thanks for pointing these issues out. We are happy to add further clarity to our draft in these areas.
>
> "The motivation of the work is unclear...": See our response to reviewer ZbFM, weakness 1. We have found that our iterative training procedure allows MAVIS to outperform PARM, reinforcing its value in the body of work on test-time multi-objective alignment for LLMs.
>
> "The novelty of the paper is limited...": We believe that the overall process of training the single-objective value models in our particular way and combining them at inference time constitutes a novel algorithm, although its individual parts borrow from prior work.
>
> "The current comparison set is narrow...": We are adding additional baselines; see our response to reviewer ZbFM.
>
> "Lack of more analysis...": We have added Figure 6 in our revised draft, which illustrates the performance change during each iteration when training value functions. We will consider adding the other analyses you brought up as time permits.
>
> "The authors claim that “MAVIS can be applied regardless of whether the weights of π_ref are available.": MAVIS decoding only requires access to the logits output by the language modeling head of the LLM, which can be provided without disclosing the weights of the model. Likewise, computing the KL divergence only requires access to these logits.
>
> "The process of training value functions may require more time compared to directly training the policy": As we note in section 4.5, the process for fully training the MAVIS value models can take a bit longer than fine-tuning a single model for each objective; thus, a baseline like Rewarded Soups may take less effort to implement. However, the performance improvement that MAVIS provides over such baselines justifies this extra effort when one needs to balance potentially conflicting objectives. On the other hand, adopting the MORLHF strategy and fine-tuning enough separate models to adequately cover the pareto front would take significantly longer than implementing MAVIS.
>
> "Additionally, the training stability remains uncertain": We have found that most training hyperparameters (e.g. learning rate, weight decay, lora rank, dropout) do not need to change across iterations or objectives, suggesting that training is relatively stable.
>
> "the performance under long output sequences has not been explored": Our experiment where we compare with PARM features a maximum completion length of 512 tokens, which is significantly longer than with the previous experiments. The success of MAVIS in that scenario suggests that it can handle long output sequences.

---

> ### Comment · Reviewer_8deh · 2025-11-28
>
> Thanks for your response. Here are some points I would like to further diccuss with authors:
>
> - Regarding the motivation section, from my view, simply comparing experimental results may not sufficiently justify the motivation behind proposing this work.
>
> - Is it currently possible to modify the logits of black-box models? Since I have not used this feature via API, the authors could further clarify this point. Additionally, the paper should explicitly explain why the method is applicable to closed-source models to avoid any misunderstanding.
>
> - Regarding the concern that "The process of training value functions may require more time compared to directly training the policy," some strong baselines might achieve better results without needing to search for weight combinations. So, authors need to do lots of experiments to show that the training of value functions indeed outperforms these baselines.
>
> - For some ongoing experiments, it would be helpful if the authors could directly update the corresponding results in the rebuttal.

---

> > ### Author Response · Authors · 2025-12-04
> > **Response to Reviewer Comment**
> >
> > "...simply comparing experimental results may not sufficiently justify the motivation behind proposing this work": We believe that the practical benefits of our method over what has been tried before, as revealed through our experiments, make it worth sharing with the research community.
> >
> > "Is it currently possible to modify the logits of black-box models?": The APIs for several popular closed-source models are able to return the log-probabilities for for the top-k candidates for some value of k. One could easily apply the MAVIS decoding method to these log-probabilities in order to obtain an aligned distribution. For OpenAI’s models the maximum value of k is too small to be practical, but for Google’s Gemini models one can access up to the top-19 probabilities which in our experience is enough for MAVIS to be useful. Google has a Colab notebook demonstrating how to access the log-probabilities from Gemini at this url: https://colab.research.google.com/github/GoogleCloudPlatform/generative-ai/blob/main/gemini/logprobs/intro_logprobs.ipynb#scrollTo=ESdZtc-uesuZ
> >
> > "...some strong baselines might achieve better results without needing to search for weight combinations": Our intention is to show that given an arbitrary generative model, MAVIS does a better job of improving that model's alignment towards arbitrary weighted combinations of objectives than other multi-objective alignment algorithms. If your concern is that there exists a model which has been aligned very well to a specific weighted combinations of objectives, that is not the kind of comparison we are interested in because that model is not producing an entire pareto front like MAVIS does. Indeed, we found that MORLHF fine-tuning a separate model for each weighting does outperform MAVIS at some points, but to produce a full pareto front requires far more training time than MAVIS. Furthermore, MAVIS could be used to enhance an already strong generative model, so the existence of high-quality models does not render it obsolete.

---

### Official Review · Reviewer_vozQ · 2025-10-30

**Soundness:** 3
**Presentation:** 2
**Contribution:** 2
**Rating:** 4
**Confidence:** 3

**Summary:**

The paper *“MAVIS: Multi-Objective Alignment via Value-Guided Inference-Time Search”* tackles the problem of multi-objective alignment during decoding. Instead of the common *weighted-logit* formulation used in prior work such as MOD, the authors propose to perform **inference-time policy search** guided by a **weighted value function**. The key novelty is that the value function is *reward-based but includes a KL regularization term*, encouraging both objective satisfaction and adherence to the base model distribution. This leads to a more balanced trade-off between competing objectives without retraining or fine-tuning the LLM. Experimental results across diverse datasets, baselines, and language models show that MAVIS achieves superior Pareto fronts and improved controllability, with convincing ablation studies that isolate the contribution of each design choice.

**Strengths:**

- **Comprehensive empirical evaluation:** The experiments span multiple datasets, baseline methods, and backbone LLMs, clearly demonstrating that MAVIS consistently outperforms previous approaches and yields clean, interpretable Pareto fronts.
- **Strong ablation studies:** The ablations effectively demonstrate how each design choice—value weighting, KL-regularized reward, and inference-time policy search—contributes to the observed improvements.
- **Intuitive motivation:** The explanation of why MAVIS outperforms MOD is clear and intuitive, highlighting how weighting the *value function* instead of *logits* better aligns with downstream rewards.
- **Principled extension:** The approach refines prior decoding-time alignment methods with subtle yet meaningful changes that improve both interpretability and empirical performance.

**Weaknesses:**

- **Limited theoretical novelty:** The theoretical modifications appear incremental. Earlier methods, such as Satisficing Alignment (Chehade et al., 2025), have also explored weighting value functions rather than logits.
- **KL-based value functions are not new:** The idea of including KL regularization in the value formulation has been previously studied, for example in (Zhou et al., 2025).
- **Positioning relative to MOD:** The introductory section does not sufficiently distinguish MAVIS from the closely related MOD framework (Shi et al., NeurIPS 2024). Clarifying the conceptual differences early would improve readability.
- **Analytical justification:** The paper would benefit from stronger theoretical grounding or complexity analysis of the mentioned bandit-style derivation. Currently, the explanation of computational infeasibility is vague.

### References

- Shi, Ruizhe, et al. *"Decoding-time language model alignment with multiple objectives."* NeurIPS 37 (2024): 48875–48920.
- Chehade, Mohamad, et al. *"Bounded Rationality for LLMs: Satisficing Alignment at Inference-Time."* arXiv preprint arXiv:2505.23729 (2025).
- Zhou, Jin Peng, Kaiwen Wang, Jonathan Chang, Zhaolin Gao, Nathan Kallus, Kilian Q. Weinberger, Kianté Brantley, and Wen Sun. *"q♯: Provably Optimal Distributional RL for LLM Post-Training."* arXiv preprint arXiv:2502.20548 (2025).

**Questions:**

1. Why prefer this specific weighting of value functions over similar approaches such as Satisficing Alignment (Chehade et al., 2025) that also operate on value-based objectives?
2. How computationally demanding is learning or estimating the value function in MAVIS?
3. Equation (5) is described as “computationally infeasible” — could you provide an analytical interpretation or quantification of this infeasibility?
4. Lines 84–86 mention that MOD requires fine-tuning large language models. Does MOD actually modify the model parameters, or does it remain a purely inference-time method?
5. How does the performance of MAVIS change when the KL-based term is removed, i.e., when using a reward-only value function?

### References

- Chehade, Mohamad, et al. *"Bounded Rationality for LLMs: Satisficing Alignment at Inference-Time."* arXiv preprint arXiv:2505.23729 (2025).

---

> ### Author Response · Authors · 2025-12-03
> **Author Rebuttal**
>
> **"Limited theoretical novelty"**: We have shown the novelty and value of MAVIS compared to other methods (including the Satisficing Alignment paper) in our responses to the other reviewers
>
> **"KL-based value functions are not new"**: While the objective for value model training in Q# is somewhat similar to that of MAVIS, Q# only considers the single-objective case and uses a different training methodology that focuses on distributional RL.
>
> **"Positioning relative to MOD"**: We elaborate on how MAVIS differs from the MOD framework in the related works and Section 3.3.
>
> **"Analytical justification"**: We do not feel there is any need for a theoretical analysis of the computational infeasibility of tilting the probability distribution over sequences in the bandit formulation; the search space grows exponentially with the maximum sequence length.
>
> **Q1**: See our response to Q2 from reviewer YrgS
>
> **Q2**: Our method for learning the value function does involve generating a large amount of tokens up-front for training the value models. For example, training the humor value model required generating under 1.5 million tokens from the reference policy for iteration 0 (which is shared across objectives), and then under 500,000 tokens from the MAVIS policy for iteration 1. However, the data generation process can be easily parallelized by running on multiple GPUs simultaneously and/or splitting a GPU into multiple instances using NVIDIA Multi-Instance GPU. The actual training of the value model is not very computationally expensive since a relatively small LLM backbone is suitable and there is no need to scale up the value model when the size of the generative model changes.
>
> **Q3**: See response to "Analytical justification".
>
> **Q4**: MOD does modify the model parameters. It actually requires a separate round of fine-tuning for each objective.
>
> **Q5**: The KL-based term in the loss for the value function and the parameter $\beta$ in the decoding expression serve a similar purpose of controlling how much KL divergence to permit in order to achieve a certain reward. The parameter $\beta$ is sufficient for constraining the KL divergence and can be tuned easily without any retraining. On the other hand, the KL penalty enables the value function to adapt based on the expected KL divergence of the future sequence, potentially allowing for a better tradeoff between the final reward and KL divergence. Thus, while MAVIS would still work without that KL penalty, its performance for the same amount of KL divergence may be a bit weaker.

---

### Official Review · Reviewer_ZbFM · 2025-11-01

**Soundness:** 2
**Presentation:** 3
**Contribution:** 2
**Rating:** 2
**Confidence:** 5

**Summary:**

This paper introduces MAVIS, an inference-time alignment method that enables dynamic multi-objective control of large language models without fine-tuning. The approach trains small value models for each objective and combines them at inference time using user-specified weights to guide token selection. Experimental results on preference datasets show that MAVIS matches or exceeds the performance of baseline methods that require fine-tuning separate models for each objective, while offering greater flexibility and computational efficiency for multi-objective alignment tasks.

**Strengths:**

The paper addresses a meaningful problem by enabling a base model to adapt to different user preferences at inference time without extensive retraining, which provides significant practical value for deploying language models that can flexibly satisfy diverse user requirements.

**Weaknesses:**

1. A significant limitation is the insufficient motivation, as the core idea of using weighted multi-objective reward models or value functions to influence model logits during decoding for preference-aligned generation has been explored in prior work with similar approaches (e.g., PARM: Multi-Objective Test-Time Alignment via Preference-Aware Autoregressive Reward Model). The authors should better articulate what distinguishes their motivation and approach at a fundamental level from existing methods.

2. The experimental evaluation is insufficient as it lacks comparison with recent state-of-the-art baselines

   [1] Rewards-in-Context: Multi-objective Alignment of Foundation Models with Dynamic Preference Adjustment

   [2] Arithmetic Control of LLMs for Diverse User Preferences: Directional Preference Alignment with Multi-Objective Rewards

**Questions:**

See weaknesses.

---

> ### Author Response · Authors · 2025-11-21
> **Author Rebuttal**
>
> Weakness 1: We thank the reviewer for pointing out this opportunity to compare with prior work. We have compared MAVIS with PARM on the safeRLHF dataset, and found that MAVIS achieves a superior pareto front when limited to the same KL divergence as PARM. The details can be found in Appendix H of our revised draft.
>
> Weakness 2: We have added Rewards-in-Context as a baseline to our helpfulness vs harmlessness plot (See Figure 3 in our revised draft), and we can add it to our other experiments as time permits.

---

> > ### Comment · Reviewer_ZbFM · 2025-11-26
> >
> > Thank you for the author's response. Although the proposed method benefits from PARM in the experiments, the core ideas are the same. Could you explain in what aspects your method provides additional advantages?

---

> > > ### Author Response · Authors · 2025-11-26
> > > **Response to Reviewer Comment**
> > >
> > > Thanks for the follow-up. One major advantage which MAVIS brings is that to introduce a new objective, you simply need to train a new value model for that particular objective while maintaining the same value models you were using before for the others. With PARM, you would have to completely retrain the value model any time you wanted to introduce a new objective. Additionally, one can see from Figure 7 that MAVIS achieves substantially higher single-objective results than PARM does, which creates a wider pareto front. We attribute this to the fact that we iteratively train each objective's value model, which as we show in our analytical results brings them closer to $Q^*$ than one can achieve by only training on sequences from $\pi^{ref}$ like PARM does.

---

### Official Review · Reviewer_YrgS · 2025-11-01

**Soundness:** 3
**Presentation:** 3
**Contribution:** 3
**Rating:** 6
**Confidence:** 3

**Summary:**

The paper introduces MAVIS — Multi-Objective Alignment via Value-Guided Inference-Time Search — a framework for aligning large language models to multiple, possibly conflicting objectives (e.g., helpfulness, harmlessness, humor) without fine-tuning. MAVIS trains small per-objective value (Q) models that estimate token-level returns under KL-regularization and combines them at inference using user-specified weights to steer decoding. The method provides a monotonic policy-improvement guarantee, integrates with beam or tree search, and empirically outperforms multi-objective fine-tuning baselines such as MORLHF, Rewarded Soups, and MOD on HH-RLHF and Summarize-from-Feedback benchmarks. It enables dynamic preference control at test-time while remaining computationally efficient compared to retraining-based alignment.

Conceptual overlap with Transfer Q★:
The core idea value-guided, KL-regularized inference-time decoding—is closely related to Transfer Q★ (Chakraborty et al., NeurIPS 2024). While MAVIS extends this to multi-objective settings with learned per-objective value models, the conceptual novelty beyond prior value-guided decoding frameworks could be clarified.

**Strengths:**

1. Introduces a principled method for multi-objective control of LLM behavior without modifying model weights — a clear step beyond single-objective inference-time methods.
2. Provides a formal derivation from KL-regularized policy optimization and proves monotonic improvement under iterative value updates, grounding the approach in RL theory.
3. Requires training only small per-objective value models instead of full fine-tuning, enabling dynamic preference mixing at inference and easy deployment on frozen LLMs.
4. Empirically Demonstrates superior Pareto fronts compared to MORLHF, Rewarded Soups, and MOD on HH-RLHF and Summarization Feedback datasets.

**Weaknesses:**

1. The evaluation focuses on numerical Pareto-front comparisons but lacks qualitative or human preference assessments. There is no demonstration of smooth or controllable trade-offs between objectives.
2. The conceptual overlap with Transfer Q★ (NeurIPS 2024).  Both rely on KL-regularized, value-guided inference-time alignment, yet the distinction in mechanism and novelty is not well-articulated.
3. The method is not generalizable, as it requires to train value model for each reward objective

**Questions:**

1. Could the authors clarify the conceptual distinction from Transfer Q★ and whether MAVIS can be interpreted as its multi-objective generalization?
2. Can you Compare Your method with --Bounded Rationality for LLMs: Satisficing Alignment at Inference-Time

---

> ### Author Response · Authors · 2025-12-02
> **Author Rebuttal**
>
> **W1**: We provide sample completions from MAVIS and the baseline methods for different tradeoffs between helpfulness and harmlessness or summary quality and faithfulness in Appendix G. The purpose of these examples is to show that the text generated is indeed aligned to a particular combination of the objectives while remaining coherent.
>
> **W2/Q1**: The Transfer-Q* paper proposes direct transfer decoding (in which you have access to a model which has already been aligned with the target reward) and indirect transfer decoding (which only requires access to a model aligned with some known reward). The reliance on existing, potentially suboptimal models allows the authors of that paper to bypass any additional fine-tuning or training of value models, but it comes with significant limitations. First, assuming the existence of a model that is already aligned with the target reward is highly impractical. Therefore, it makes more sense to compare MAVIS with indirect transfer decoding. Since that algorithm relies on importance sampling in their approximation of Q*, the resulting policy could be highly suboptimal if the underlying reward for the available model is too different from the target reward. In contrast, through MAVIS' iterative value model training we obtain a value model which adapts to the distribution shift as the MAVIS policy becomes better-aligned with the reward. Another disadvantage of using Transfer-Q* is that it introduces a much larger decoding overhead than MAVIS (this is based on results which the authors reported during their rebuttal period for NeurIPS 2024 [1], where they were ultimately accepted). In the table they provide in their general response comparing TQ* with naive decoding and other baselines, the reported inference time for TQ* is over 10 times larger than that of naive decoding. In contrast, according to our tests the decoding speed is only reduced by around 54% when using MAVIS. In conclusion, the methods used in MAVIS both differ conceptually from Transfer-Q* and enable better performance. We do not believe that MAVIS should be interpreted as a multi-objective generalization of Transfer-Q*.
>
> [1] https://openreview.net/forum?id=5PrShrKxoX&referrer=%5Bthe%20profile%20of%20Ming%20Yin%5D(%2Fprofile%3Fid%3D~Ming_Yin4)
>
> **W3**: As far as we are aware, all existing multi-objective alignment methods require more effort as the number of objectives increases, in the form of additional training and/or additional computation during inference time. With MAVIS it is possible to circumvent much of the additional inference-time computation through value model distillation, ensuring that the cost of scaling the number of objectives is only felt during training.
>
> **Q2**: The paper "Bounded Rationality for LLMs: Satisficing Alignment at Inference-Time" focuses on constrained optimization where the reward for one objective needs to be maximized while the others need to be kept above a given threshold. In contrast, the purpose of MAVIS is to extend the Pareto front as far as possible so that for any choice of objective weights, the best result can be achieved in expectation. Furthermore, since the code for that paper does not appear to be available at this time and all of their results are in terms of win-rates, it would be difficult to do a direct comparison.

---

### Author Response · Authors · 2025-11-21
**Notice of Revisions to Draft**

We thank the reviewers for their time and comments. Here we would like to mention that since making our submission, we have obtained new results for MAVIS on the experiments using the Llama 2 7B generative model which outperform the results shown in the original draft. These improvements are primarily attributed to removing the value normalization step during decoding (with $\beta$ being increased to compensate) and increasing the number of top-$k$ tokens considered. We have updated figures 3, 4, and 5 as well as Table 2 and all tables in Appendix F to reflect the new results. All other changes in our revised draft are pointed out in the individual rebuttals.

---

> ### Author Response · Authors · 2025-12-04
>
> Since our previous revision, we have been attempting to finish our implementation of the Rewards-In-Context baseline, but unfortunately due to an issue with the code we have not been able to reproduce it faithfully within the time constraints of this rebuttal period. We have have removed it from the helpfulness/harmlessness plot in Figure 3.

---

### Meta-Review · Area_Chair_wuAn · 2026-01-07

**Summary:**

This paper introduces MAVIS, a lightweight inference-time framework that steers a frozen LLM toward any user-specified mix of conflicting objectives by training small per-objective value models and combining them at decode-time with KL-regularized search. However, the submission raised significant concerns. Reviewers found the motivation weak because test-time multi-objective control has been tackled by PARM, GenARM and others; the core technical moves including value-guided decoding with KL penalty and soft policy iteration were viewed as incremental. The authors should further emphasize the advantages of their work compared with previous studies, demonstrate their unique insights, and strengthen the novelty of their work. Based on the above considerations, I recommend rejection, while encouraging the authors to further develop this promising direction.

**Reviewer Concerns:**

Partially addressed:
  - Empirical edge over PARM
  - Relationship to Transfer

Still outstanding:
  - Fundamental novelty versus GenARM/PARM/Satisficing Alignment.
  - Theoretical contribution beyond soft policy iteration + KL penalty.
  - Comprehensive SOTA baseline comparison, human evaluation, fluency/diversity metrics, cost analysis, long-sequence test.

**Reviewer Scores:**

- YrgS: keep positive
- ZbFM: keep negative
- vozQ: keep negative
- 8deh: keep negative

---

### Decision · Program_Chairs · 2026-01-26

Reject